# A multiscale approach reveals elaborate circulatory system and intermittent heartbeat in velvet worms (Onychophora)

Henry Jahn [1✉], Jörg U. Hammel [2], Torben Göpel[3,4], Christian S. Wirkner [5] & Georg Mayer [1]

An antagonistic hemolymph-muscular system is essential for soft-bodied invertebrates. Many ecdysozoans (molting animals) possess neither a heart nor a vascular or circulatory system, whereas most arthropods exhibit a well-developed circulatory system. How did this system evolve and how was it subsequently modified in panarthropod lineages? As the closest relatives of arthropods and tardigrades, onychophorans (velvet worms) represent a key group for addressing this question. We therefore analyzed the entire circulatory system of the peripatopsid *Euperipatoides rowelli* and discovered a surprisingly elaborate organization. Our findings suggest that the last common ancestor of Onychophora and Arthropoda most likely possessed an open vascular system, a posteriorly closed heart with segmental ostia, a pericardial sinus filled with nephrocytes and an impermeable pericardial septum, whereas the evolutionary origin of plical and pericardial channels is unclear. Our study further revealed an intermittent heartbeat—regular breaks of rhythmic, peristaltic contractions of the heart—in velvet worms, which might stimulate similar investigations in arthropods.

[1] Department of Zoology, Institute of Biology, University of Kassel, Heinrich-Plett-Straße 40, D–34132 Kassel, Germany. [2] Institute of Materials Physics, Helmholtz-Zentrum Hereon at DESY, Notkestraße 85, D–22607 Hamburg, Germany. [3] Multiscale Biology, Johann-Friedrich-Blumenbach Institut für Zoologie und Anthropologie, Georg-August-Universität Göttingen, Friedrich-Hund-Platz 1, D–37077 Göttingen, Germany. [4] Department of Biological Sciences, University of North Texas, 1155 Union Circle #305220, Denton, TX 76203, USA. [5] Institut für Allgemeine und Spezielle Zoologie, Institut für Biowissenschaften, Universität Rostock, Universitätsplatz 2, D–18055 Rostock, Germany. ✉email: henry.jahn@uni-kassel.de

Non-arthropod ecdysozoans, including nematodes, priapulids, kinorhynchs and allies, possess relatively simple circulatory systems, which lack pulsatile organs such as hearts or contractile vessels[1–4]. Their hemolymph rather circulates passively due to the contractions of body musculature during locomotion[5,6]. This contrasts with the elaborate circulatory system in most panarthropods (= onychophorans + tardigrades + arthropods), which typically consists of vascular and lacunar parts and involves one or more pulsatile organs[7–9]. While this holds true for onychophorans (velvet worms) and arthropods (chelicerates, myriapods, crustaceans and hexapods), tardigrades (water bears) might have lost their vascular system and pumping heart due to miniaturization[10–13]. The vascular systems, i.e., hearts and off-branching arteries of onychophorans and arthropods, are characterized by cellular linings that are missing in their lacunar systems[8,9,14]. These linings comprise non-epithelial, apolar cells[14–17] and in this respect they clearly differ from the vascular endothelium of vertebrates[18]. The body cavity of adult onychophorans comprises a hemocoel (sometimes referred to as mixocoel), which is surrounded by the extracellular matrix and arises during embryogenesis by mixocoely, i.e., a fusion of primary and secondary/coelomic body cavities[19,20].

The organization of both the vascular and the lacunar systems has been well explored in arthropods[7,8,21,22]. The vascular systems show a more or less uniform organization across the major taxa, with an ostiated contractile heart (dorsal vessel) pumping the hemolymph into the paired, serially arranged cardiac arteries that open into the body cavity[7,8,23,24]. This type is therefore referred to as an "open" vascular system, whereas that of vertebrates is considered "closed". However, there is still some debate as to whether or not the "closed" type truly exists[8,18,25–28]. In contrast to the more or less conserved vascular systems, the hemolymph pathways of the lacunar system vary greatly across the arthropod subgroups[8,22,23,29–34], correlating with their diverse body plans.

The circulatory system of onychophorans has not been characterized in as much detail as that of arthropods, although this would be necessary for understanding the evolutionary changes that have taken place in the arthropod and panarthropod lineages. Like in arthropods, the vascular system of onychophorans comprises a tubular dorsal heart equipped with paired segmental ostia, but there is no indication of associated cardiac arteries[5,9,14]. The anterior and posterior extents of the heart as well as the question of whether its posterior end is open or closed are still unclear[5,14,35–38]. Physiological studies revealed myogenic control of heart pulse, the rate of which is affected by specific drugs[39]. It was further deduced from principles of fluid dynamics known from arthropods that the onychophoran heart might pump the hemolymph anteriorly[5,40–42], but neither its contraction rhythm nor the direction of hemolymph flow have been measured. The position and orientation of cardiac ostia also await clarification in onychophorans. From different arthropod taxa it is known that these slit-like openings of the heart wall either correspond in position with legs or are shifted anteriorly, posteriorly or contralaterally, or they are completely missing in specific body segments[8,21]. Furthermore, ostia may be vertically oriented or diagonally tilted in arthropods[8,21], whereas their specific arrangement is unknown in onychophorans.

The two dorsal antennal arteries seem to be the only true vessels (defined by distinct tissue linings; see Table 1 and Supplementary Table 1 for terminology and definitions) in the onychophoran body besides the heart[41]. Their spatial and compositional relationship to the so-called "supracerebral sinus"[41] remains obscure. Specifically, the heart and the antennal arteries seem to be separated from each other by this sinus, which would suggest that their linings are disrupted. This would further imply that the vascular system of onychophorans does not comprise a unitary structure or functional unit, which seems unlikely and remains to be clarified. Alternatively, the tissue that lines the "supracerebral sinus" might have been overlooked in onychophorans, which is indeed indicated by the report of a "septum" separating this hemal space from the remaining body cavity[41]. Hence, it is unclear whether the structure described as "supracerebral sinus" is a true vessel lined with cells or rather a sinus belonging to the lacunar system. Furthermore, it cannot be ruled out currently whether or not other vessels might have been overlooked in onychophorans.

Even more controversy surrounds the composition of other parts of the lacunar system in these animals. The lacunar system of the onychophoran trunk comprises a large perivisceral sinus (surrounding the gut), a dorsal pericardial sinus (surrounding the heart), a pair of lateral longitudinal sinuses (harboring the ventral nerve cords, the salivary glands and the nephridia), and numerous transverse channels associated with the annuli or plicae of the integument[5,9,40,43]. These transverse channels have been variously referred to as "hemal channels"[44–46], "lacunae"[35], "vascular channels"[43,44,47], "subcutaneous lateral channels"[47], "lacunar blood spaces" and components of the "intermuscular channel system" ("lakunäre Bluträume" and "intermuskuläres Kanalsystem" *sensu* Gaffron[9]). They might be either an autapomorphy of Onychophora or a symplesiomorphy inherited from the last common ancestor of Panarthropoda. Manton & Heatley[5] and Gaffron[9] reported that these channels are dorsally associated with the pericardial sinus, but beyond this to our knowledge nothing is known about their extent and distribution. Likewise, the pericardial sinus has been assumed to be confluent with the perivisceral sinus via perforations in the pericardial septum[5,9], but the existence of such perforations has not been convincingly shown.

In summary, only little is known about the pathways of hemolymph through the onychophoran body. This and other open issues regarding the organization of the onychophoran circulatory system hamper evolutionary conclusions based on comparisons with arthropods. While the homology of the onychophoran heart with that of arthropods has been established[10,21], the evolutionary relationship of the other elements of their circulatory systems remains obscure. Unfortunately, paleontological data from lobopodians, which most likely comprise a non-monophyletic assemblage of stem lineage representatives of tardigrades, onychophorans, arthropods, and panarthropods as a whole, respectively[48–50], are not helpful for resolving these issues due to incomplete preservation and highly conjectural reconstructions of the circulatory system in these fossils[21,51–54]. It seems therefore crucial to obtain details on the organization and spatial relationship of vessels, sinuses and hemal channels in onychophorans. To achieve this goal, we analyzed the circulatory system of the Australian velvet worm species *Euperipatoides rowelli* (Peripatopsidae) by combining histology, histochemistry, confocal laser scanning microscopy (CLSM), scanning electron microscopy (SEM), synchrotron radiation-based X-ray micro-computed tomography (SR-µCT), three-dimensional (3D) reconstructions, and video recordings. The obtained results contribute to a better understanding of the functional anatomy of the circulatory system in velvet worms and the last common ancestor of Onychophora and Arthropoda, which inhabited the Earth over ~561 million years ago[55].

## Results
We used a multiscale approach to analyze the circulatory system of the onychophoran *E. rowelli* by combining histology, histochemistry, synchrotron radiation-based X-ray micro-computed tomography, scanning electron microscopy, confocal microscopy,

**Table 1 Specific terminology for components of circulatory system and associated tissues and cells in onychophorans.**

| Suggested term | Reference(s) for suggested term | Figure | Description | Synonym(s) | References for synonyms |
|---|---|---|---|---|---|
| Antennal ring channel* | – | 2e, f; 3d; 9b; S3a–c | Transverse subcutaneous →channel associated with individual annuli of antennal integument | – | – |
| Antennal artery | 96 | 3a, b, d–h; 9a, b; S3a–c | Dorsal longitudinal →vessel extending from antennal basis to antennal tip; it originates laterally from →supracerebral region of anterior aorta and opens distally into →ventral antennal channel | Antennal vessel | 41 |
| Anterior aorta* | – | 2a, b; 3a–d; 9a, b; S2; S8; S9a–c | Cephalic →vessel, which connects →heart to →antennal arteries; it is composed of anterior →supracerebral region of anterior aorta and posterior →suprapharyngeal region of anterior aorta | Supracerebral sinus | 41 |
| Cardiac valve* | – | 9a, b; S1b, c | Ventral valve at the anterior terminus of →heart; it opens into →anterior aorta | Aortic valve / Anterior heart valve | 21 / 134 |
| Dorsomedian channel* | – | 6d, g; 9b; c; S4c | Short connection between →plical channel and →pericardial sinus situated dorsal to →heart | Rautenförmige Lücken [German] | 9 |
| Head cavity* | – | 3d; 9b; S1a–f | Part of →lacunar system located within head | Head hemocoel | 41 |
| Heart | 5,9,10,14,36,37,39,103 | 2a, b; 7a, e; 9; S5; S7a, b | Contractile dorsal longitudinal →vessel, which is lined with cardiomyocytes and perforated with segmental →ostia; it opens anteriorly into →suprapharyngeal region of anterior aorta | Dorsal vessel / Herz / Rückengefäß [German] / Rückengefäss, / Rückenkanal [German] / Спинной сосуд [Russian] | 41 / 16 / 9,135 / 136 / 136 / 36,37 |
| Hemocyte | 14,103,137 | 6f; 7e; S3; S7a | Individual cell of →circulatory system, which is either sessile (attached to tissues) or suspended in →hemolymph | – | – |
| Lateral channel* | – | 2g, h; 5a, b | Longitudinal hemolymph space within →lateral sinus, which connects main leg cavities of one body side | Lateral haemocoel | 5 |
| Lateral sinus | 138,139 | S10a | Ventrolateral compartment of main body cavity, which is delimited by →transverse septum and ventrolateral body wall; it contains salivary glands, nephridia and ventral | Lateral compartment / Längskanal [German] | 103,140 / 9 |

**Table 1 (continued)**

| Suggested term | Reference(s) for suggested term | Figure | Description | Synonym(s) | References for synonyms |
|---|---|---|---|---|---|
| | | | nerve cords; →**lateral channel** is part of lateral sinus | - | - |
| Leg cavity | 56 | 2g, h; 4b–f | Main cavity of leg, which is subdivided into four compartments by sheets of septal muscles and is associated with additional →**lacunae**, including foot cavity and →**leg ring channel** | - | - |
| Leg ring channel* | - | 5b, d–f; 9b, c | Transverse subcutaneous →**channel** associated with individual annuli of leg integument | Transverse haemal ring channel | 56 |
| Nephrocyte | 57,104 | 6e; 9a; S5d; S6e | Pinocytotic, stationary, podocyte-like cell within →**lacunar system;** nephrocytes occur either in clusters or as individual cells | Cellules à carminate [French]<br>Néphrocytes à carminate [French]<br>Large multinuclear pericardial cell<br>Pericardial cell<br>Pericardiazelle [German]<br>Small uninucleate pericardial nephrocyte | 141<br>107<br>5<br>5<br>142<br>5 |
| Ostium (plural: ostia) | 9 | S5d; S6b, d | Slit-like perforation of heart wall, which is associated with valves and connects →**pericardial sinus** with lumen of →**heart** | Herzspalten [German] | 9 |
| Pericardial cells | 5 | S4f; S6e | Small, uncharacterized cells, which together with →**nephrocytes** form longitudinal bands on either side of →**heart** within →**pericardial sinus** | - | - |
| Pericardial channel* | - | 2c, d; 6a, c, g; 7c, d; 9a–c; S4d, e; S7b | Segmental channel situated between dorsolateral muscles of body wall and ventrolateral extension of →**pericardial septum**; each pericardial channel is located at level of →**ostia** and legs and links →**perivisceral cavity** with →**pericardial sinus** | - | - |
| Pericardial conglomerate* | - | 7a; 9a; S4f; S5a; S6e | Longitudinal bands of cells on either side of →**heart** within →**pericardial sinus,** which include clusters of →**nephrocytes** and smaller, uncharacterized pericardial cells | Fat body<br>Pericardial network<br>Fatty tissue | 143<br>144<br>5 |
| Pericardial septum | 9,103 | 6c; 7a; S4d–f; S5a, c, d; S11 | Horizontal →**septum,** which is situated dorsal to digestive tract and is attached to ventral wall of →**heart;** it forms ventral border of →**pericardial sinus** | Pericardial floor | 5 |

**Table 1 (continued)**

| Suggested term | Reference(s) for suggested term | Figure | Description | Synonym(s) | References for synonyms |
|---|---|---|---|---|---|
| Pericardial sinus | 14,103 | 2c, d; 3a, b; 5a; 6g; 9b, c | Sinus between dorsolateral longitudinal muscles and →**pericardial septum**, which contains →**heart** and →**pericardial conglomerate**; it receives hemolymph (via posterior region and →**pericardial channels**) from →**perivisceral sinus** and (via →**dorsomedian channels**) from →**plical channels** and directs it via →**ostia** into lumen of →**heart** | Pericardial cavity<br>Perkardialsinus [German]<br>Perikardialhöhle [German]<br>Sinus péricardique [French] | 145<br>57<br>146<br>147 |
| Perivisceral sinus | 103 | 2g, h; 3d; 5a, b; 7 | Central compartment of main body cavity, which contains digestive tract, slime glands, genital tract and accessory genital glands; it is delimited dorsally by →**pericardial septum** and laterally by →**transverse septa**; it is confluent via perforations in →**transverse septa** with →**lateral sinuses** and communicates via →**pericardial channels** with →**pericardial sinus** | Median compartment<br>Средний отдел [Russian]<br>Perivisceral haemocoel<br>Central compartment | 103<br>36,37<br>5<br>140 |
| Plical channel* | - | 2e, f; 3d; 5a; 6a, d, f, g; 7c; 9b, c; S4a–c; S7a, b | Subcutaneous, mostly ring-like hemal →**channel** associated with each transverse fold (=plica) of body integument; like plicae, plical channels have rather irregular shape and arrangement in region of head and anal cone | Hemal channel<br>Lacunae<br>Lakunäre Bluträume [German]<br>Intermuskuläres Lakunensystem [German]<br>Subcutaneous lateral channel<br>Vascular channel | 44,46<br>35<br>9<br>9<br>47<br>43,44,47 |
| Spinous pad channel* | - | 5c, d | Transverse subcutaneous →**channel** underlying each spinous pad situated ventrally in distal part of leg | - | - |
| Supracerebral region of anterior aorta* | - | 3a, b; 9a, b; S2a; S9b, c | Anterior part of →**anterior aorta**, which is located dorsal to brain and links →**suprapharyngeal region of anterior aorta** with →**antennal arteries** | - | - |
| Suprapharyngeal region of anterior aorta* | - | 3a, b; 9a, b; S2e–h; S8; S9a | Posterior part of →**anterior aorta**, which is situated dorsal to pharynx and connects →**heart** to →**supracerebral region of anterior aorta** | - | - |

**Table 1 (continued)**

| Suggested term | Reference(s) for suggested term | Figure | Description | Synonym(s) | References for synonyms |
|---|---|---|---|---|---|
| Transverse septum | 148 | 5a, b; 9b, c | Bilateral transverse →**septum**, which is formed by dorsoventral musculature; it separates →**periviceral sinus** from both →**lateral sinuses**, although all three sinuses remain confluent with each other via slit-like perforations in transverse septum | Inner giant<br>circular muscle<br>Transversal musculature<br>Septum transversal [French]<br>Поперечные мускулы [Russian] | 43<br>103<br>147<br>36,37 |
| Ventral antennal channel* | - | 3d–h; S3 | Ventral longitudinal →**channel**, which extends from antennal tip to antennal basis; it receives distally hemolymph from →**antennal artery** and directs it proximally into main →**head cavity**; ventral antennal channel is confluent with →**antennal ring channels** | Antennal hemocoel | 41 |

Asterisks (*) indicate novel terms. Terms that are highlighted in bold and preceded by an arrow represent references to the corresponding terms in the first column of Table 1 and Supplementary Table 1.

3D reconstruction, and video recordings. The circulatory system of *E. rowelli* is represented by vascular and lacunar portions, which differ from each other in that the vascular system consists of true vessels lined with (nonepithelial) cells whereas the lacunar system comprises an elaborate complex arrangement of channels, lacunae and sinuses without any dedicated tissue lining (Figs. 1–9, Table 1; Supplementary Figs. S1–S11, Supplementary Movies 1–7, Supplementary Table 1). Although the two systems are functionally confluent, yet morphologically distinguishable from each other, they are treated separately in the following.

**Organization of the vascular system and intermittent heartbeat.** The vascular system of *E. rowelli* occupies by volume only ~1% (Supplementary Table 2) of the entire circulatory system of the animal and consists of four main elements: (i) the heart (HRT); (ii) the suprapharyngeal region of the anterior aorta (ATA, SPR); (iii) the supracerebral region of the anterior aorta (SCR); and (iv) a pair of antennal arteries (ANA, Figs. 2a, b, 3a, b). The suprapharyngeal and supracerebral regions are subunits of the anterior aorta. The above mentioned four elements form a functional unit and are clearly lined with a continuous cellular layer (Fig. 3c; Supplementary Figs. S1a–f, S3c, d, S4f, S8, S9). The heart is an elongated, tubular structure (⌀ 150–200 µm) situated mid-dorsally between the two dorsal bundles of longitudinal muscles and extending from the first to the ultimate (15th) leg-bearing segments (Fig. 2a, b; Supplementary Figs. S1, S2, S5, Supplementary Movie 1). It occupies only 0.36% of the entire volume of the circulatory system (Supplementary Table 2). A valve with paired lateral lips formed by the dorsal cardiac wall represents the anterior limit of the heart at the boundary to the suprapharyngeal region of the anterior aorta (Fig. 9a, b; Supplementary Fig. S1a–h). While the heart opens anteriorly into the suprapharyngeal region of the anterior aorta, it bears a blind posterior end in the 15th leg-bearing segment (Figs. 3a, b, 7e; Supplementary Data 1). The heart is

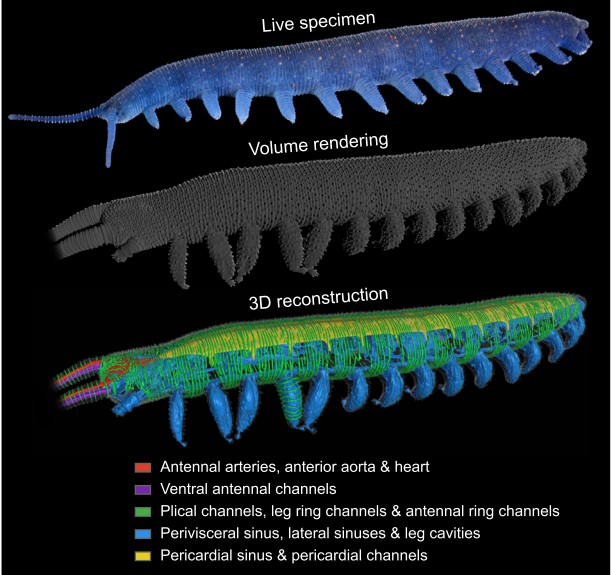

**Fig. 1 Habitus and gross morphology of the circulatory system of *E. rowelli*.** Top: Photograph of living specimen (~5 cm in length). Middle: Volume rendering of SR-µCT dataset from ~2.3 cm long specimen. Bottom: 3D reconstruction of the circulatory system of same specimen. Note that plical channels were segmented automatically based on grey value range. Gaps in these channels represent artifacts due to clusters of hemocytes with bright grey values. Leg ring channels are illustrated only for the fourth leg.

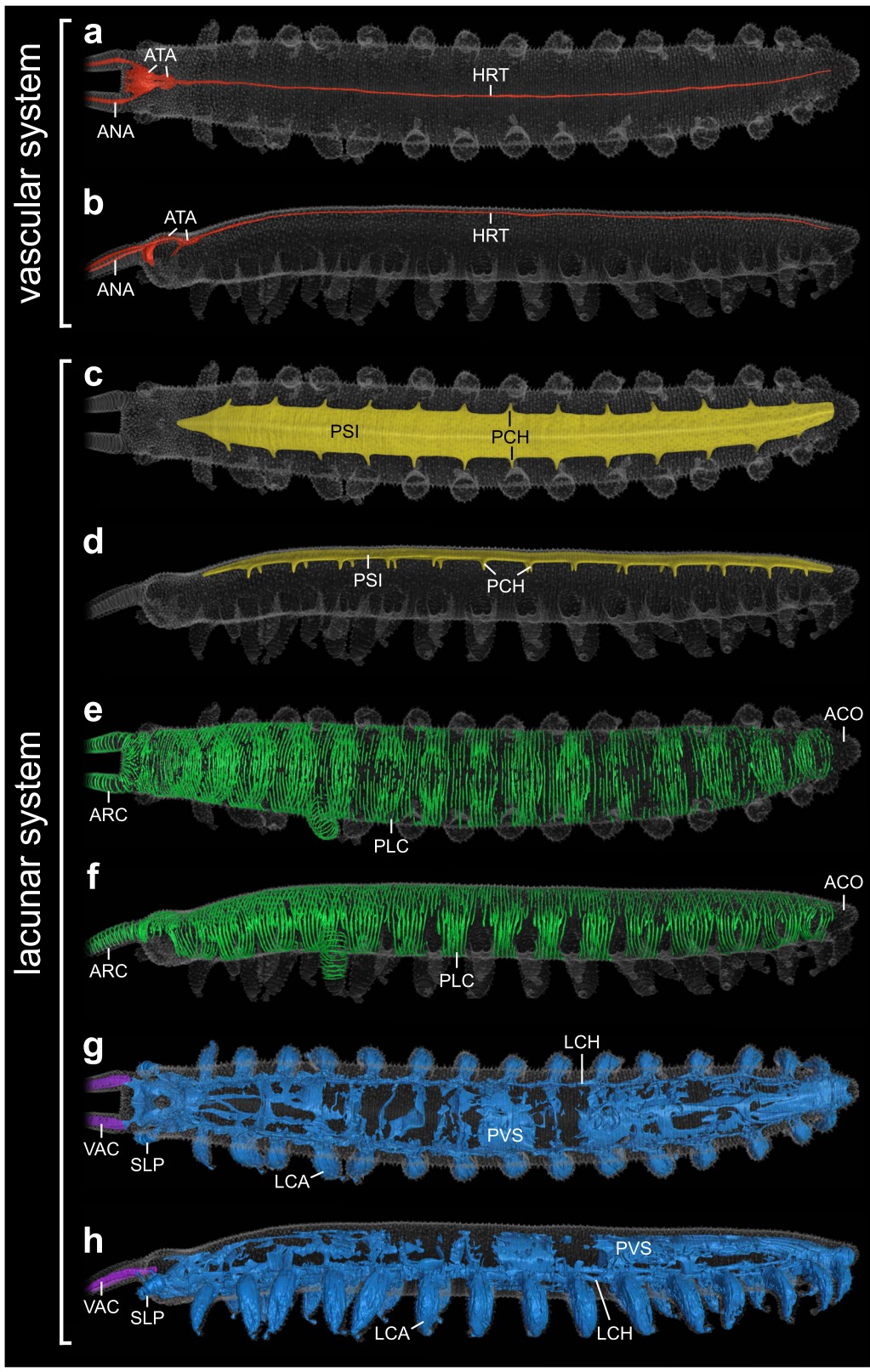

innervated by an unpaired dorsomedian nerve, which bulges into the cardiac lumen along the length of the heart (Fig. 7e; Supplementary Fig. S4c, f).

The heart wall consists of a thin layer of cardiomyocytes arranged in a semicircular circumvolution pattern without a helical shift (Fig. 6c; Supplementary Fig. S5g, h). The ventral wall of the heart tube adheres directly to the pericardial septum (PSE

in Figs. 6b, c, 7a, e; Supplementary Figs. S1g, h, S5a, c–e), whereas ligaments of connective tissue link its dorsal wall to the two dorsal bundles of longitudinal muscles belonging to the body wall (Fig. 6e; Supplementary Fig. S5b). Paired, segmental ostia (OST) perforate the dorsolateral walls of the heart; each pair of ostia corresponds in position with pericardial channels (PCH) and legs of the respective segment (Figs. 4a, b, 7a, d, 9c; Supplementary

**Fig. 2 Overview of vascular and lacunar systems of *E. rowelli*.** 3D reconstructions of individual components of circulatory system based on SR-µCT data combined with semi-transparent body outline. Dorsal (**a**, **c**, **e**, **g**) and lateral views (**b**, **d**, **f**, **h**). Anterior is left in all images. Only proximal part of antennae is shown. **a**, **b** Components of vascular system including heart, anterior aorta and antennal arteries. **c**, **d** Pericardial sinus with segmental pericardial channels corresponding in position with legs. Note that pericardial channels are missing in first and last leg-bearing segments. **e**, **f** Plical channels, antennal ring channels and leg ring channels (illustrated only for the fourth left leg). Note absence of plical channels in anal cone. **g**, **h** Ventral antennal channels and remaining part of lacunar system surrounding nervous and muscular systems as well as digestive, excretory, secretory and reproductive organs. Note fractured appearance of periviscereal sinus. Note also that lateral channels are confluent with leg cavities. Abbreviations: ACO anal cone, ANA antennal artery, ARC antennal ring channel, ATA anterior aorta, HRT heart, LCA leg cavity, LCH lateral channel, PCH pericardial channel, PLC plical channel, PSI pericardial sinus, PVS periviscereal sinus, SLP slime papilla, VAC ventral antennal channel.

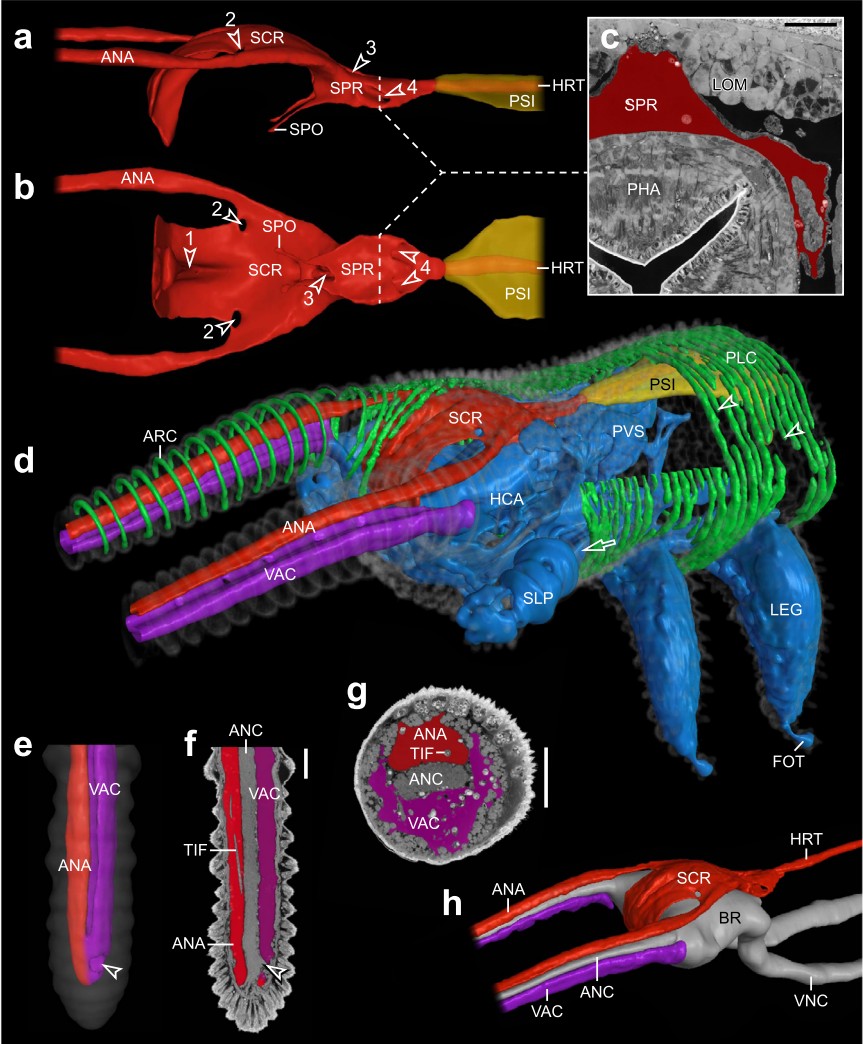

**Fig. 3 Organization of circulatory system in head and antennae of *E. rowelli*.** 3D reconstructions based on SR-µCT data (**a**, **b**, **d**, **e**, **h**) and micrographs of semi-thin sections (**c**, **f**, **g**). Dorsal is up in **a**, **c,d**, **g**, **h** and left in **e**, **f**. Ventral view in **b**. **a**, **b** Details of anterior aorta. Arrowheads indicate position of muscles and nerves: 1, frontal muscle; 2, oral lip nerves; 3, tongue muscle; and 4, pharyngeal protractor muscles. **c** Cross section (inverted grey) of suprapharyngeal region of anterior aorta (pseudo-colored in red). **d** Overview of cephalic circulatory system. Arrowheads point to incomplete plical channels in pedal region; arrow indicates channel-like connection between main head cavity and cavity of slime papilla. **e–g** Spatial relationship of antennal artery (pseudo-colored in red), antennal nerve cord, and antennal channel (pseudo-colored in purple). Note nerve fibers (arrowheads) crossing circulatory system in antennal tip. **h** Overview illustrating spatial relationship of major components of cephalic circulatory system and central nervous system. Abbreviations: ANA antennal artery, ANC antennal nerve cord, ARC antennal ring channel, BR brain, FOT foot, HCA head cavity, HRT heart, LEG leg, LOM longitudinal musculature of body wall, PHA pharynx, PLC plical channels, PSI pericardial sinus, PVS periviscereal sinus, SCR supracerebral region of anterior aorta, SLP slime papilla, SPO ventral opening of suprapharyngeal region of anterior aorta, SPR suprapharyngeal region of anterior aorta, TIF uncharacterized tissue fiber, VAC ventral antennal channel, VNC ventral nerve cord. Scale bars: 100 µm (**c**, **f**, **g**).

Figs. S5d, e, g, S6a–e). The individual ostia appear as narrow slit-like openings equipped with a double-flap valve projecting deep into the lumen of the heart (Fig. 4b; Supplementary Figs. S5d, e, g, 6b, d). The valves exhibit strong signal in samples labeled for

F-actin, revealing a transverse, dorsolateral arrangement of muscle fibers within each valve (Supplementary Fig. S6c).

Video footage from mid-body segments of a dissected specimen reveals an autonomous rhythmic contraction of the heart

(Fig. 8a, b; Supplementary Data 2; Supplementary Movies 2, 7). During systole, the diameter of the heart is reduced to about half of the most dilated condition during diastole (Fig. 8a–d, Supplementary Data 2). Measurements from two regions situated 1.5 mm apart (blue and red squares in Fig. 8a, b) revealed a constant contraction rhythm with short pauses. The average contraction cycle took ~1.6 seconds and the average contraction rate was ~34.4 heart beats per minute. Several consecutive contractions were interrupted by peculiar, prolonged breaks lasting for 3.4 ± 2.1 seconds, so that the average beat rate between two pauses was slightly higher at ~39.5 beats per minute (Fig. 8b). The selected anterior region of the heart (blue curve in Fig. 8c) contracted 168 ms later than the posterior region (red curve in Fig. 8c). This implies an anterograde peristaltic contraction at a speed of 8.9 mm per second. Overlays of 14 successive contractions exhibit a highly congruent pattern, which can be broken down into four major phases, including contraction (phase I), contraction peak (phase II), expansion (phase III), and maximum expansion plateau (phase IV). Notably, phase I is shorter and its slope is steeper than that of phase III (Fig. 8d). The anterograde flow of the hemolymph is confirmed by the anterior motion of hemocytes within the heart lumen (Supplementary Movie 2). After a fast-forward flow of hemolymph during the systole, it subsequently slows down and the hemocytes indicate a short backflow during the diastole (Supplementary Movies 2, 3). The forward movement of the hemocytes during the systole is rapid, so that they are hardly visible at the original speed of the recording, whereas their posterior movement during the diastole is considerably slower and extends over a much shorter distance. The opening and closing mechanism of ostial valves follows the heart contraction rhythm, in that the valves close during systole and open during diastole (Supplementary Movie 3).

We further calculated basic cardiac physiological properties on the basis of mean values of five dimension measurements from light microscopy and scanning electron microscopy data (*end diastolic radius* ($r_{max}$) = 80 μm; *end systolic radius* ($r_{min}$) = 40 μm; *length* ($l$) = 18 mm) and heart rate measurements ($HR$ = 34.4 beats min$^{-1}$) of video recordings. The heart volume was calculated for two different conditions: the end diastolic volume (*EDV* at $r_{max}$) and the end systolic volume (*ESV* at $r_{min}$).

$$EDV = \pi * r_{max}^2 * l = \pi * (0.08\,\text{mm})^2 * 18\,\text{mm} = 362\,\mu m^3 \quad (1)$$

$$ESV = \pi * r_{min}^2 * l = \pi * (0.04\,\text{mm})^2 * 18\,\text{mm} = 90.5\,\mu m^3 \quad (2)$$

Based on these measurements, the stroke volume ($SV = EDV - ESV$), is 272 μm$^3$, the ejection fraction ($EF = \frac{SV}{EDV}$) is 75%, and the cardiac output ($CO = SV * HR$) is 9.357 mm$^3$ min$^{-1}$. Furthermore, *EF* and *HR* allow for an estimation of the full circulation time (i.e., the time it takes for the entire hemolymph volume to be circulated once) based on the volume of the entire circulatory system (*CV*) and the volume occupied by the heart (*EDV*). According to the measurements, it takes 10 min and 45 seconds for the entire hemolymph volume to be circulated.

$$Relative\,flow\,rate = \frac{EDV}{CV} * EF * HR = 0.36\% * 75\% * 34.4\,\text{min}^{-1} = 9.3\%\,\text{min}^{-1} \quad (3)$$

$$Circulation\,time = \frac{100\%}{Relative\,flow\,rate} = \frac{100\%}{9.3\%\,\text{min}^{-1}} = 10:45\,\text{min} \quad (4)$$

From the heart, the hemolymph is then pumped anteriorly through the cardiac valve (CAV) into the anterior aorta, which is a relatively wide but flat vascular structure linking the heart lumen with the two antennal arteries (Figs. 2a, b, 3; Supplementary Figs. S1b, c, S2b, d). The anterior aorta is subdivided into a relatively small posterior suprapharyngeal region situated above the pharynx and a larger anterior supracerebral region located dorsal and anterior to the brain (Figs. 2a, b, 3a–d; Supplementary Figs. S1, S2, S8, S9, Supplementary Movie 5). The suprapharyngeal region of the anterior aorta has no ventral lining but is instead bordered ventrally by the pharynx itself, whereas its lateral and dorsal regions are delimited by connective tissue, which adheres to the lateral walls of the pharynx (Fig. 3c; Supplementary Figs. S1f, S2g, h). Its median part is voluminous and vessel-like, whereas the lateral parts are rather flat (Fig. 3a–c). Paired ventral openings situated lateral to the pharynx direct the hemolymph from the suprapharyngeal region of the anterior aorta into the main cavity of the head (SPO in Fig. 3a, b; arrowhead in Supplementary Fig. S2e).

Anteriorly, the suprapharyngeal region of the anterior aorta joins the supracerebral region of the anterior aorta (Fig. 3a, b). At the transition zone between the two aortic regions, the connective tissue of their lateral lining is attached medially to the tongue muscle (arrows in Supplementary Fig. S2a, f). The supracerebral region of the anterior aorta encloses the dorsal part of the brain (Fig. 3a, b, h). Like the suprapharyngeal region of the anterior aorta, the supracerebral region of the anterior aorta has no ventral lining but rather borders directly the dorsal neurilemma of the brain, whereas its lateral walls consist of strands of connective tissue that extend from the brain to the dorsal body wall (arrows in Supplementary Fig. S2b, d). The anterior aorta is pierced by the paired pharyngeal protractor muscles, the first pair of oral lip nerves and the unpaired tongue muscle (Fig. 3a–c; Supplementary Figs. S2a, f–h, S8, S9a, c, Supplementary Movie 5). Additionally, a prominent frontal muscle bulges into the anteromedian portion of the anterior aorta and thus causes a keel-like indentation in its 3D-reconstructed shape (Fig. 3b, d, h; Supplementary Figs. S1i, S2c, d). The anterior aorta opens anteroventrally into the lacunar system of the head, whereas its lateral branches lead into the two dorsal antennal arteries where the lateral lining of connective tissue reduces gradually and fuses with the contact zone of the brain and the dorsal musculature as well as the eyes (Figs. 2a, 3a, b, d, h, 9; Supplementary Figs. S2b, d, S9).

Each antennal artery appears as a tubular structure situated dorsal to the antennal nerve cord and extending to the antennal tip where it opens into the antennal channel situated ventral to the antennal nerve cord (Fig. 3a, b, d–h; Supplementary Fig. S3). Sets of fibers project from the antennal nerve cord and fan out distally in the region where the antennal artery fuses with the antennal channel (Fig. 3e, f; Supplementary Fig. S3b). Like other components of the vascular system, the antennal arteries are lined with a distinct layer of connective tissue; in contrast to the ventral antennal channels (VAC), they are not connected to the ring channels associated with the annuli of antennal integument (Supplementary Fig. S3a–d).

**Organization of the lacunar system**. The lacunar system of *E. rowelli* constitutes nearly 99% of the circulatory system and consists of various sinuses and lacunae (Supplementary Table 2). The head contains several spaces and channels associated with the antennae and slime papillae as well as prominent cavities surrounding the brain and the pharynx with all their extensions (Fig. 3d–h; Supplementary Figs. S3, S8, S9a–c). Intricate subcutaneous plical channels (PLC) also belong to the lacunar system of the head (Figs. 2e, f, 3d). While the antennae possess ring channels associated with the annuli of the antennal integument, the jaws and the slime papillae do not show such channels due to the lack of annulation. The subcutaneous ring channels of each antenna are associated with the ventral longitudinal antennal channel (Supplementary Fig. S3). In contrast to the tube-like antennal artery, the antennal channel is crescent-shaped in cross section (Fig. 3d, g; Supplementary Fig. S3). It receives

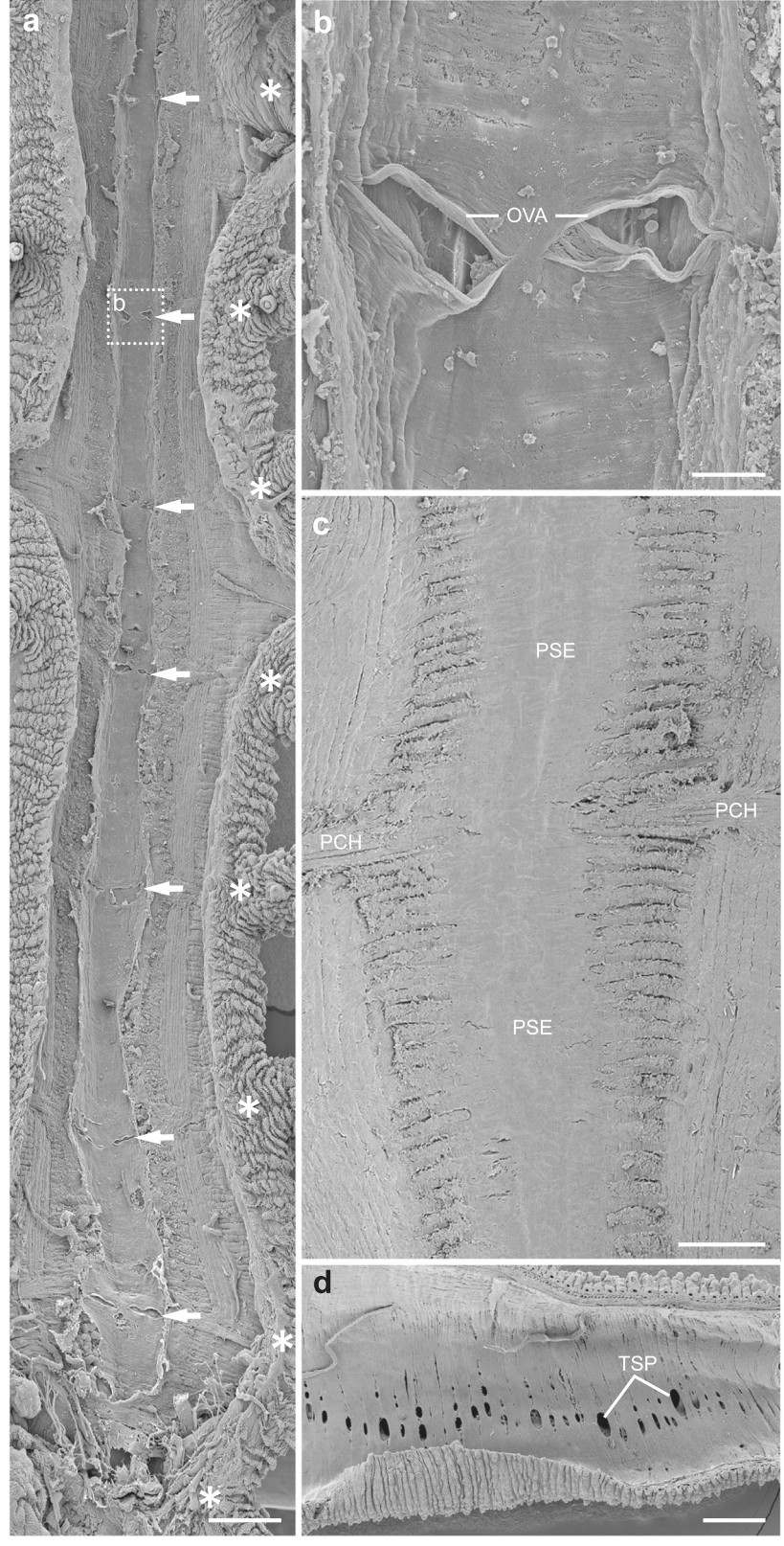

**Fig. 4 Overview and details of heart, ostia, pericardial sinus, pericardial septum, pericardial channels and transverse septa in dissected specimens of** ***E. rowelli.*** Scanning electron micrographs (**a**–**d**). Ventral (**a**–**c**) and lateral (**d**) views. Anterior is up in **a**–**c** and left in **d**. **a** Overview of dissected specimen. Note correlation in position of ostia (arrows) with legs (asterisks). **b** Detail of bilateral ostial valves in dissected heart. **c** Detail of pericardial septum, which shows compact and continuous organization without perforations. Striated pattern is due to transverse muscle fibers inside pericardial septum. **d** Detail of transverse septum. Note numerous slits and openings that link perivisceral sinus with lateral sinus. Abbreviations: PCH pericardial channel, PSE pericardial septum, OVA ostial valves, TSP slit-like perforations in transverse septum. Scale bars: 300 μm (**a**, **c**), 50 μm (**b**), 400 μm (**d**).

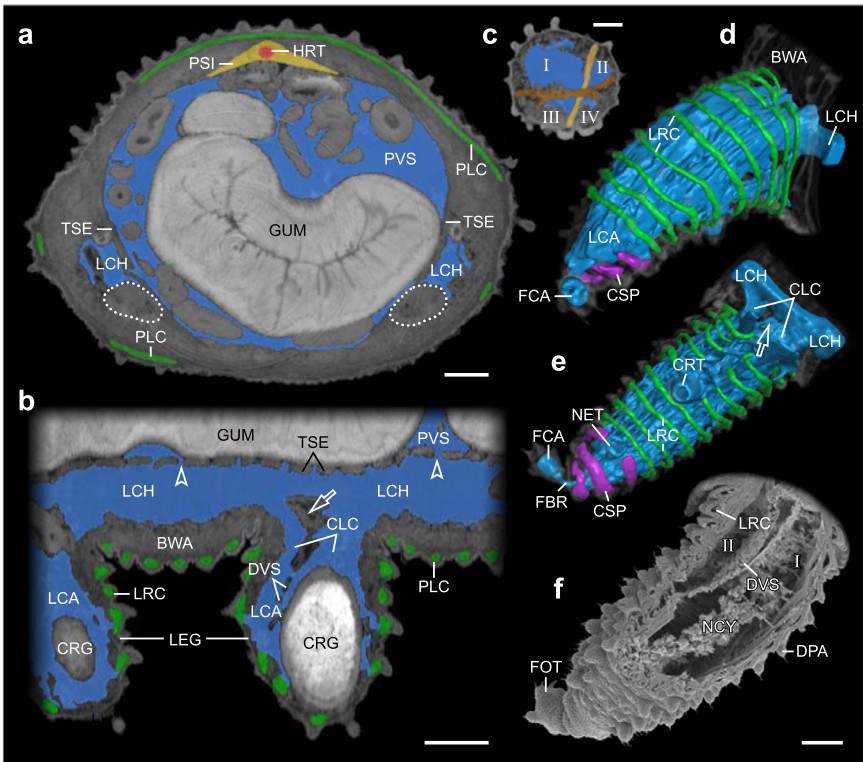

**Fig. 5 Organization of circulatory system in trunk and legs of *E. rowelli*.** Virtual sections from SR-μCT image stack (**a**, **b**, **c**), 3D reconstruction of SR-μCT data (**d**, **e**) and scanning electron micrograph (**f**). Dorsal is up in **a**, **c**; anterior is left in **b**; proximal is in upper right corner in **d**–**f**. **a** Pseudo-colored cross section of trunk showing spatial relationship of heart (red), pericardial sinus (yellow), plical channels (green), and perivisceral and lateral sinuses (blue). Dotted lines indicate ventral nerve cords. **b** Pseudo-colored horizontal section of mid-trunk region with legs. Note that lateral channel is confluent with main leg cavity. Arrow points to leg depressor muscle (cf. Oliveira et al.[56]). Arrowheads indicate slit-like perforations in transverse septum linking perivisceral and lateral sinuses. **c** Pseudo-colored cross section of leg showing spatial relationship of four major compartments of leg cavity (cf. Oliveira et al.[56]) separated from each other via anteroposterior septal muscle (brown) and dorsoventral septal muscle (light brown). **d**–**f** Circulatory system of leg. Main leg cavity (blue) extends via narrow bridge into foot and is associated with several leg ring channels (green) and spinous pad channels (purple). **d** Posterodorsal view of fourth left leg. **e** Ventral view of same leg as in **d**. Arrow points to space occupied by leg depressor muscle that crosses channel connecting main leg cavity with lateral channel. **f** Partially dissected leg in dorsal view. Note dorsoventral septal muscle subdividing main leg cavity into anterior and posterior compartments and prominent cluster of nephrocytes inside anterior compartment of leg cavity. Abbreviations: BWA body wall, CLC connection of main leg cavity and lateral channel, CRG crural gland, CRT crural tubercle, CSP spinous pad channel, DPA dermal papilla, DVS dorsoventral septal muscle of leg, FBR foot bridge, FCA foot cavity, FOT foot, GUM midgut, HRT heart, LCA leg cavity, LCH lateral channel, LEG leg, LRC leg ring channel, NCY nephrocytes, NET nephridial tubercle, PLC plical channel, PSI pericardial sinus, PVS perivisceral sinus, TSE transverse septum. Scale bars: 400 μm (**a**, **b**), 200 μm (**c**, **f**).

hemolymph from the antennal artery at the antennal tip and passes it further into the main cavity of the head (HCA in Figs. 3d–f, 9a, b). This cavity comprises a complex arrangement of lacunae and interstitial spaces, some of which extend into the jaws and slime papillae. The cavity of each slime papilla is linked with the main cavity of the head via a channel (arrow in Fig. 3d).

The lacunar system of the trunk is confluent with that of the head and differs from it in that the trunk is subdivided by several septa and muscle sheets into four major sinuses: (i) the pericardial sinus (PSI) surrounding the heart; (ii) the perivisceral sinus (PVS) harboring the digestive and reproductive tracts, the slime glands, the gonad/s, the anterior and posterior accessory genital glands; and (iii and iv) the two lateral sinuses that contain the ventral nerve cords, nephridia, and salivary and crural glands (Fig. 5a, b; Supplementary Figs. S7, S10). The pericardial sinus is dorsally delimited by the dorsolateral longitudinal muscles of the body wall and ventrally by the pericardial septum, which is attached to the dorsolateral body wall (Supplementary Figs. S4f, S5a). Our data, including complete series of histological and semi-thin sections as well as scanning electron micrographs and SR-μCT scans, revealed that the pericardial septum is a non-perforated,

continuous tissue layer that isolates the pericardial sinus from the perivisceral sinus (Figs. 4c, 7a; Supplementary Figs. S4f, S11). However, the pericardial sinus opens posteriorly, specifically at the level of the posterior end of the heart, into the perivisceral sinus (arrow in Fig. 7e; Supplementary Data 1). Moreover, paired segmental pericardial channels link the pericardial sinus with the perivisceral sinus (Figs. 2c, d, 4c, 6a, c, g, 7d; Supplementary Fig. S11). These segmental channels are formed by ventrolateral extensions of the pericardial septum and correspond in position with ostia (Figs. 2c, d, 6c, 7a; Supplementary Fig. S11).

The perivisceral sinus and the two lateral sinuses are amorphous and voluminous cavities that harbor most organs of the trunk and are separated from each other by a transverse septum (=dorsoventral musculature, TSE). However, they are still confluent with each other via numerous slits and openings within the transverse septum (Fig. 4d; Supplementary Fig. S11). The perivisceral sinus runs along the entire trunk and is confluent posteriorly with the pericardial sinus (Figs. 2g, h, 7e). The lateral sinuses form two longitudinal cavities (=lateral channels, LCH) on either side of the perivisceral sinus (Figs. 2g, h, 5a, b; Supplementary Figs. S7a, S10a,b; Supplementary Movie 4). The

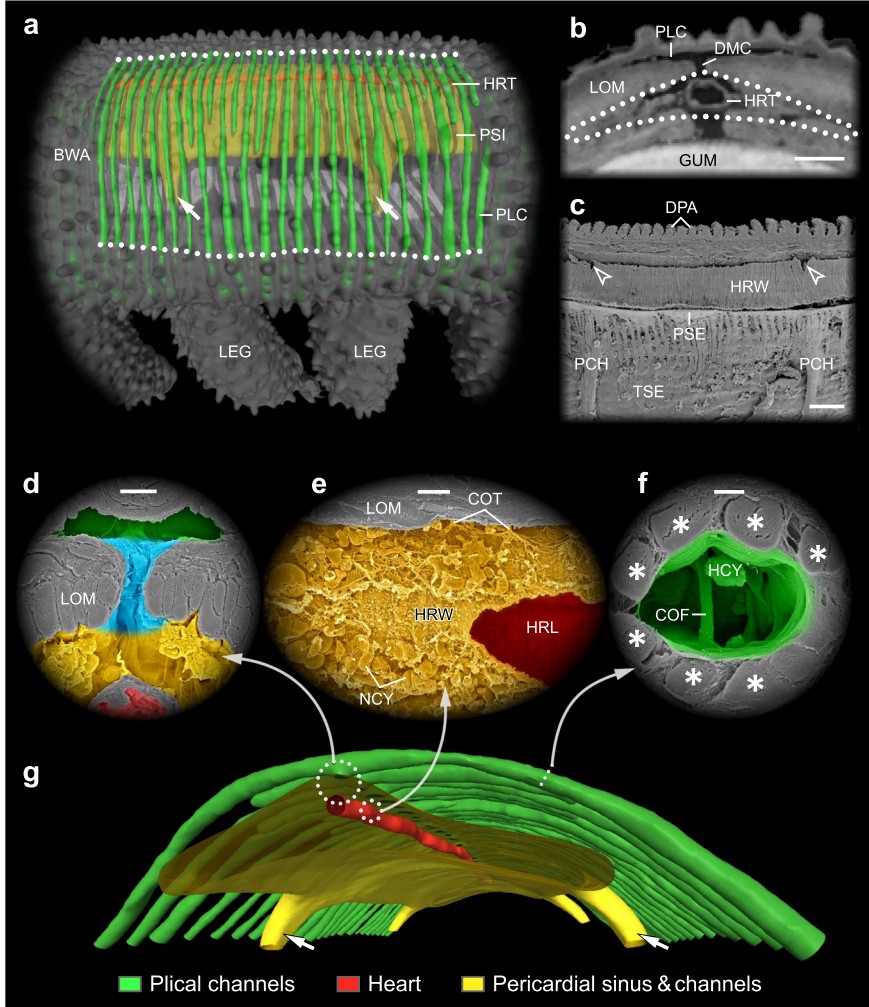

**Fig. 6 Channels and structures associated with heart and pericardial sinus of *E. rowelli*.** 3D reconstructions (**a**, **g**), virtual cross section from SR-μCT image stack (**b**) and scanning electron micrographs (**c**–**f**, pseudo-colored in **d**–**f**). Dorsal is up in all images. **a** Spatial relationship of heart (red), pericardial sinus (yellow) and plical channels (green) in two trunk segments in dorsolateral view. Dotted lines indicate virtually dissected dorsolateral body wall. Arrows point to pericardial channels. **b** Spatial relationship of pericardial sinus (dotted line), heart, dorsomedian channel and plical channel in dorsal body region. **c** Sagittal section illustrating correspondent segmental arrangement of ostia (arrowheads) and pericardial channels. **d** Detail of dorsomedian channel (cyan) linking plical channel (green) with pericardial sinus (yellow). Heart lumen in red. **e** Oblique section of outer (yellow) and inner heart wall (red). **f** Detail of plical channel (green) crossed by collagen fibers and surrounded by bundles of circular musculature (asterisks). **g** Spatial relationship of plical channels, heart, pericardial sinus, and segmental pericardial channels (arrows). Abbreviations: BWA body wall, COF collagen fibers, COT connective tissue attaching heart to dorsolateral longitudinal muscle, DMC dorsomedian channel, DPA dermal papillae, GUM midgut, HCY hemocyte, HRL lumen of heart, HRT heart, HRW heart wall, LEG leg, LOM longitudinal musculature of body wall, NCY nephrocyte, PCH pericardial channel, PLC plical channel, PSE pericardial septum, PSI pericardial sinus, TSE transverse septum. Scale bars: 100 μm (**b**), 150 μm (**c**), 15 μm (**d**), 25 μm (**e**), 10 μm (**f**).

lateral channels of each body side are confluent with the individual main leg cavities via a wide opening, which is crossed by two dorsoventral proximal leg muscles (Fig. 5b, f). Each main leg cavity (LCA) is subdivided by two nearly perpendicular sheets of septal muscles into four compartments that fuse with each other in the distal leg region (Fig. 5c, f; see Oliveira et al.[56] for further details). The cavity of the distal leg region is, in turn, linked via a narrow channel of the foot bridge with the cavity of the foot (Fig. 5d, e).

The plical channels are the outermost components of the lacunar system (Figs. 1, 2e, f, 3d, 6a, 9c). Like the other constituents of the lacunar system, they are not lined with cells but only with extracellular matrix (Figs. 6f, 7c; Supplementary Fig. S4a–c). Plical channels occupy nearly 9% the volume of the entire circulatory system, i.e., more than twentyfold the volume of the heart lumen (Supplementary Table 2). They comprise a series of ring-like channels (15 per segment) situated within the plicae

of the integument, more specifically, embedded between bundles of the circular musculature of the body wall (Figs. 5a, b, 6a, d, f, g; Supplementary Fig. S4a–c). Each plical channel is surrounded by 5–10 bundles of circular muscle fibers extending along the channel (Figs. 6f, 7c; Supplementary Fig. S4a–c). Many plical channels form almost complete rings that are confluent with the two lateral channels (Fig. 2e, f; Supplementary Movie 6), whereas some plical channels, especially those in the leg-bearing regions, end blindly in the lateral body region (Figs. 3d, 6a; Supplementary Movie 6). Each plical channel of the trunk dorsally joins via a short dorsomedian channel (DMC) into the pericardial sinus (Fig. 6b, d, g; Supplementary Figs. S4c, S5d). Close to this region, bundles of collagen fibers cross the lumen of plical channels (Fig. 6f; Supplementary Fig. S4a–c).

Like the plical channels of the trunk, the subcutaneous ring channels of antennae and legs form series of rings corresponding to the integumental folds on the surface of each appendage

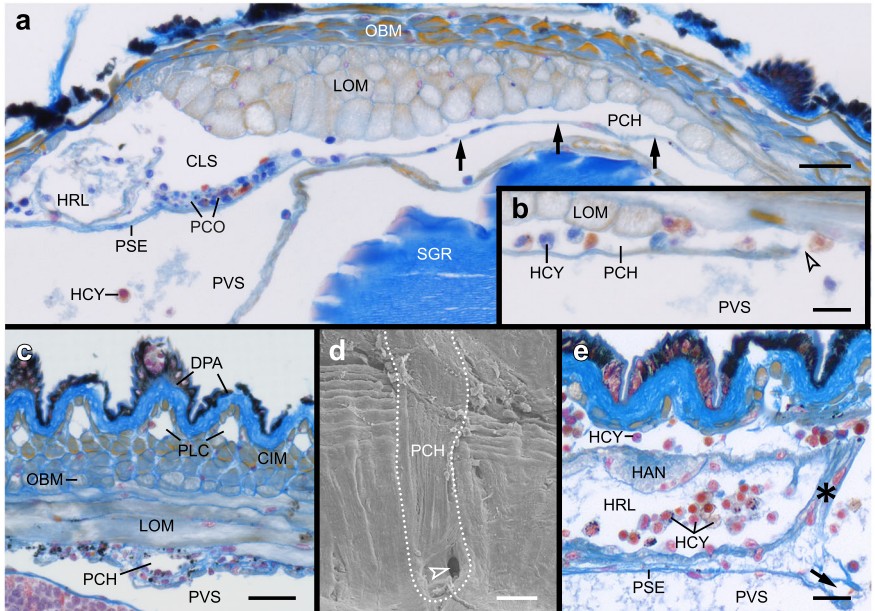

**Fig. 7 Details of lacunar and vascular circulatory system in trunk of *E. rowelli*.** Histological cross sections (**a**, **b**), horizontal section (**c**), sagittal section (**e**), and scanning electron micrograph (**d**). Dorsolateral is up in **a**, **b**; dorsal is up in **c**–**e**. **a** Overview of pericardial channel. Arrows point to ventrolateral extension of pericardial septum, which separates pericardial channel from perivisceral sinus. **b** Detail showing opening of pericardial channel (arrowhead) into perivisceral sinus. **c** Detail of subcutaneous plical channels and pericardial channel. Note that both are widely separated by musculature of body wall. **d** Surface view of pericardial channel and its opening into perivisceral sinus (arrowhead). **e** Medio-sagittal section through posterior terminus of heart, which is blindly closed (asterisk). Abbreviations: CIM circular musculature of body wall, CLS channel-like space between pericardial conglomerate, DPA dermal papillae, HAN heart nerve, HCY hemocyte, HRL lumen of heart, LOM longitudinal musculature of body wall, OBM oblique musculature of body wall, PCH pericardial channel, PCO pericardial conglomerate, PLC plical channels, PSE pericardial septum, PVS perivisceral sinus, SGR reservoir of slime gland. Scale bars: 25 μm (**a**, **e**), 10 μm (**b**), 50 μm (**c**, **d**).

(Figs. 1, 3d, 5d, e, 9b, c). The leg ring channels (LRC) are confluent with the main cavity of the leg (Fig. 5b, d, e), whereas the antennal ring channels (ARC) open into the ventral antennal channel (Supplementary Fig. S3).

**Nephrocytes, hemocytes and other cells**. We recognized three major cell types that are associated with the circulatory system of *E. rowelli*: the hemocytes (HCY), the nephrocytes (NCY), and other, uncharacterized pericardial cells (PCE). The hemocytes are solitary cells that are either sessile (attached to various tissues) or float freely in the hemolymph and appear in various parts of the circulatory system, including the heart, the anterior aorta, the plical channels, the antennal arteries and channels, the cavities of legs and feet as well as the lateral, perivisceral and pericardial sinuses (Figs. 6f, 7a, b, e; Supplementary Figs. S1, S3, S4a, Supplementary Fig. S7a). In contrast to hemocytes, the nephrocytes appear as large cells with faintly stained cytoplasm in histological sections (cf. methylene blue staining in Supplementary Figs. S5d, S6e). They occur exclusively in stationary clusters of 6–8 podocyte-like epithelial cells that enclose a blind intercellular cavity; the nephrocytes of each cluster are connected by desmosomes and rest on a basal lamina, which surrounds the entire cluster (inset in Fig. 9a; cf. Seifert & Rosenberg[57]). We detected such clusters in two regions of the circulatory system including the pericardial sinus and the anterodorsal compartment of each main leg cavity (Figs. 5f, 6e; Supplementary Figs. S4f, S5a–d; cf. Oliveira et al.[56]). Most clusters of nephrocytes are found in the pericardial sinus, in which they are arranged in two ventrolateral longitudinal bands (=pericardial conglomerate, PCO) on either side of the heart. These bands are interrupted, however, by channel-like segmental spaces (labeled as CLS in Supplementary Fig. S6e) that link the pericardial channels with the ostia of the heart. Between the clusters of large nephrocytes, other, smaller cells of different sizes that are not further characterized occur in the pericardial sinus (Fig. 6e; Supplementary Figs. S5c, d, S6e). The nephrocytes and these other, uncharacterized pericardial cells are closely associated with tracheal tubes that supply the wall of the heart and the pericardial septum (Supplementary Figs. S4f, S6d, e).

## Discussion

As soft-bodied, multi-legged invertebrates, onychophorans rely on an efficient hemolymphatic and hemodynamic system. This system not only serves as a hydrostatic skeleton for locomotion[40,58,59] but also plays roles in respiration[60], nutrition[5,61], hormone dispersion[62–64], immune response[65,66], and excretion[5,57,67]. Its origin most likely dates back to the early Cambrian (~520 Mya), i.e., the time when an onychophoran-like body plan first appeared[48,68–70].

The emergence of the circulatory system, along with specialized limb musculature, was key to successful radiation of panarthropods[56,71,72]. Our knowledge of its ancestral composition has been hampered by the limited information from crucial taxa, such as Onychophora. We therefore used a multi-methodological approach to explore the organization of the circulatory system and cardiac physiology in the velvet worm *E. rowelli*, one of the best-studied onychophoran species[17]. The new findings help to resolve several controversies surrounding the composition and operational principles of the circulatory system in onychophorans and provide insights into its organization in the last common ancestor of Onychophora and Arthropoda.

Neither scanning electron microscopy nor histology or μCT data from *E. rowelli* confirm previously reported slit-like perforations in the pericardial septum of onychophorans[5,9] that

would allow an influx of hemolymph into the pericardial sinus. The pericardial septum of *E. rowelli* instead represents a dense sheet of tissue, which seals off the pericardial sinus from the perivisceral sinus. The only interaction between the two sinuses occurs via the segmental pericardial channels and a posterior opening in the pericardial septum. The dense and compact pericardial septum of *E. rowelli* contrasts with the fenestrated dorsal diaphragm, which separates the pericardial sinus from the perivisceral sinus in most arthropods[7,8].

There has also been disagreement regarding the open[14,35–38] versus closed nature[5] of the posterior terminus of the onychophoran heart. Our analysis of serial histological sagittal sections and 3D reconstructions revealed that the heart of *E. rowelli* is indeed posteriorly closed. While some arthropods (e.g., pycnogonids, xiphosurans, some myriapods and non-malacostracan as well as some malacostracan crustaceans and most hexapods) also possess a posteriorly closed heart tube[21,23,29,38,73–75], others (such as arachnids, some myriapods, most malacostracans, and "basal-branching" hexapods) exhibit an unpaired posterior aorta emanating from the heart[21,22,74,76–81]. The absence of a posterior aorta in onychophorans supports the assumption that this structure might have arisen several times independently in arthropods[21], whereas the segmental cardiac arteries that are attached to the heart wall might be an autapomorphy of Arthropoda[8,21]. These findings indicate that the posterior heart tube ended blindly in the last common ancestor of Onychophora and Arthropoda. The posterior connection between the pericardial and perivisceral sinuses, which has been retained in extant onychophorans, might have been an anatomical constraint compensating for the limited exchange of hemolymph in this body region due to the posteriorly closed heart and the lack of a posterior aorta.

In contrast to the posterior terminus of the onychophoran heart, its anterior terminus is open. However, it has been unclear so far whether it opens directly into the main cavity of the head[5,35] or rather into a large "supracerebral sinus", also referred to as "aorta" or "enlarged anterior end of the dorsal vessel"[41]. Our data suggest that the heart of *E. rowelli* releases hemolymph into the anterior aorta via an anteroventral slit, the cardiac valve, which most likely prevents a reflux of hemolymph back into the heart. The cardiac valve of *E. rowelli* is thus reminiscent of the paired laterally suspended lips of mandibulates ("aortic valve" *sensu* Göpel and Wirkner[21]). Therefore, this arrangement might have been present in the last common ancestor of Onychophora and Arthropoda, whereas the valve with a single dorsally suspended lip might represent an autapomorphy of Chelicerata[21].

The anterior aorta of *E. rowelli* is subdivided into a suprapharyngeal and a supracerebral region. The dorsal and lateral parts of the supracerebral region of the anterior aorta are lined with connective tissue, whereas in the ventral region its lumen directly borders the brain. Based on the continuity equation of fluid dynamics[58], the relatively voluminous supracerebral region of the anterior aorta might decelerate the hemolymph flow in the cephalic region in order to provide sufficient time for an exchange of compounds between the hemolymph and cerebral tissue. A structural "blood-brain barrier" restricting paracellular diffusion, as present in some arthropods[82], however, is lacking in onychophorans[27]. The suprapharyngeal region of the anterior aorta encompasses the anteriormost tip of the heart tube and receives hemolymph from the heart, whereas the supracerebral region of the anterior aorta forwards the hemolymph frontally into the two antennal arteries and the main cephalic cavity (Fig. 9a, b). Due to their morphological and positional correspondences, the anterior aortas of onychophorans and arthropods might be homologous structures. On the one hand, they clearly show a vascular composition (for arthropods, see

references[8,21,41,83]). On the other hand, the anterior aortas in both groups are associated with the anterior part of the heart and the cardiac/aortic valve. The position of the anterior aorta relative to the brain is also different in both groups. In arthropods, it passes the brain ventrally, whereas it does so dorsally in onychophorans[41]. Thus, an anterior aorta might have been present in the last common ancestor of Onychophora and Arthropoda, but its shape and relative position to the brain might have changed in either lineage. This is not surprising, given the extensive and most likely independent reorganizations of the head during cephalization processes in different panarthropod lineages[69,84–88].

While the anterior aortas of onychophorans and arthropods are likely homologous[8,41], the afferent dorsal vessel and the efferent ventral channel associated with the protocerebral antenna of onychophorans is unlikely to be homologous with the circulatory system of the deutocerebral antenna of mandibulates, as these specialized cephalic appendages belong to different head segments[69,84,87,89,90]. Whether or not the antennal artery, the ventral antennal channel and the antennal ring channels associated with the onychophoran antennae were also present in the last common ancestor of Onychophora and Arthropoda is unclear, as the organization of the circulatory system in the anterior appendages of Cambrian outgroup taxa, such as †*Aysheaia pedunculata*[91], †*Onychodictyon ferox*[69] and †*Antennacanthopodia gracilis*[45], is unknown.

The new findings further confirm a hemodynamic connection between the heart and the two antennal arteries in onychophorans as well as a unidirectional flow of hemolymph within each antenna, which returns back into the head via ventral antennal channels ("antennal haemocoel" *sensu* Pass[41]). This network might provide a sufficient supply of hemolymph to the antennal tip, which contains a high number of chemoreceptors[92]. We additionally detected peculiar bilateral, longitudinal strands of tissue that extend from the connective tissue lining of the anterior aorta to the antennal tips by traversing the lumen of each antennal artery. The cellular composition and function of these tissue strands are unknown.

The plical channels of onychophorans, which have been variously called "hemal channels"[44–46], "lacunae"[35], "vascular channels"[43,44,47], "subcutaneous lateral channels"[47], "lacunar blood spaces" and "intermuscular channel system" (German: "lakunäre Bluträume" and "intermuskuläres Kanalsystem" *sensu* Gaffron[9]), might fulfil a supportive function as a hydrostatic skeleton for the plicae, in addition to supplying the integument and the dermal papillae with hemolymph. It should be noted that the serially arranged plical channels of onychophorans do not correspond to the cardiac arteries of arthropods[8,21,24], as they have different composition, show opposite directions of hemolymph flow and open in different body regions. Structures corresponding to the plical channels of onychophorans might well have supported each annulus in extinct lobopodians, given their distinct body annulation[44,50,69,91,93]. However, again, due to the lack of fossil evidence and the absence of such channels in extant tardigrades[94], we can only speculate about their existence in the last common ancestor of Panarthropoda.

Our results revealed that the cardiac ostia of *E. rowelli* are paired, transverse, segmental dorsolateral slits with double-flap valves protruding deep into the heart. This organization is similar to that in arachnids, myriapods and insects, which may indicate functional constraints during evolution associated with terrestrialization[8,21]. The ostia correlate in position with the pericardial channels, as both are present in the second to fourteenth leg-bearing segments but are missing in the first and last leg-bearing segments. In agreement with previous reports[5,9], we detected no branches or veins associated with the onychophoran

heart. The dorsolateral position of ostia further contrasts with the reported dorsal arrangement in the neotropical peripatid *Epiperipatus edwardsii*[9] and the South African peripatopsid *Peripatopsis capensis*[9,35]. Whether these deviating observations are due to species-specific differences or preparation artefacts remains to be clarified.

We did not notice an anterior or posterior shift in position of ostia within each segment of *E. rowelli*, as these are rather aligned with legs. Each ostium links the pericardial sinus to the lumen of the heart and is equipped with a double-flap valve system, which prevents systolic backflow of hemolymph into the pericardial sinus. Labeling for F-actin revealed prominent muscle fibers inside each flap that might be responsible for an active closure of ostia during systole. This is evidenced by the transverse, dorsoventral arrangement of muscle fibers, the contraction of which would thicken the flaps, thus pressing them against each other and eventually closing the ostial slit. This construction would require a synchronous contraction of cells in ostial valves and adjacent heart wall. We therefore anticipate specialized neurons associated with the dorsal heart nerve[14,16,95] in corresponding regions that might control these contractions. Neuroanatomical studies should aim at identifying these neurons.

Quantitative measurements revealed that heart contractions occur periodically in *E. rowelli* in anteriorly directed peristaltic waves. The in situ measured heart rate is in line with the results of physiological studies in other species, such as *Eoperipatus weldoni*[39] and *Peripatopsis* sp.[96], except that in *E. rowelli* we additionally observed regular breaks after several successive rounds of contraction. These intermittent contractions might have remained unnoticed in onychophorans because their detection requires a prolonged monitoring of heart rate. To our knowledge, comparable intermittent breaks in cardiomyocyte contraction have been reported only from animals with heartbeat reversals (bidirectional switchover), such as ascidians and insects[7,97–100]. Although the role of intermittent heartbeat remains unknown in *E. rowelli*, as this species shows unidirectional heartbeat, we speculate that the detected intermission intervals might be periods of rest that allow the cardiomyocytes to recover. It would be interesting to know whether intermittent heartbeat occurs in arthropods other than insects, as this would help to clarify whether or not it was present in the last common ancestor of Onychophora and Arthropoda.

The consistency and regularity of intermittent contractions of the heart also require further investigation, in particular the impact of drugs that have been shown to affect the rate of heartbeat in onychophorans[39] and various insects[7,101]. While we are aware that immobilization and dissection of specimens and application of saline might alter the hemolymph pressure as well as the physiological state of the individual, the observed regular heartbeat in *E. rowelli* and its striking correspondences to the results from two other onychophoran species based on entirely different datasets and methodologies (heart beat rate measurements, electrocardiograms and intracellular recordings[39,96]) indicate that these measurements might be close to the natural condition. All these studies indicate that the heart nerve might control the rhythmicity of contractions autonomously. However, we cannot rule out that there might be also an input from the central nervous system in intact specimens, as all three studies involved dissection, removal of hemolymph components, and disruption of hemolymph pressure, which might have affected heart function.

The hemolymph cells floating in the heart lumen showed fast forward and slow backward movements, similar to the grasshopper *Schistocerca americana*[102]. Based on these live and anatomical observations as well as quantitative measurements, we propose that the heart of *E. rowelli* sucks in the hemolymph via

the 13 pairs of ostia (in trunk segments 2–14) during diastole and then pumps it into the anterior aorta during systole (Fig. 8). The observed anterior flow direction corresponds to that in most arthropods[7,8] and was therefore most likely present in the last common ancestor of Onychophora and Arthropoda.

The present study revealed an unexpectedly complex organization of the circulatory system in *E. rowelli*, which comprises a combination of segmental and non-segmental features as well as vascular and non-vascular elements. For example, the ostia and pericardial channels exhibit segmental arrangement, whereas the heart, the anterior aorta and the antennal arteries are the only vascular structures. The vascular system, thus, constitutes a relatively small portion (~1%) of the onychophoran circulatory system. All parts of the lacunar system of *E. rowelli*, including various lacunae, channels and sinuses, are fluid-dynamically connected to the heart lumen. This organization represents a typical open vascular system[8], in which organs and tissues are bathed with the hemolymph supplying oxygen and nutrients and removing waste matter.

Anatomical reconstructions and measurements of hemolymph flow allow us to resolve the major circuit of the onychophoran circulatory system (Fig. 9a–c). During systole, the heart pumps the hemolymph anteriorly into the suprapharyngeal region of the anterior aorta, from which it passes into the supracerebral region of the anterior aorta and the two dorsal antennal arteries (Fig. 9a, b). The ostia and the cardiac valve ensure overall unidirectional circulation. In the distal part of each antennal artery, the hemolymph flows into the two ventral antennal channels that are associated with serially arranged antennal ring channels and proximally join the main cephalic cavity (Fig. 9b). This cavity additionally receives hemolymph (via a pair of medioventral suprapharyngeal channels) from the suprapharyngeal region of the anterior aorta and (via a wide frontal sinus) from the supracerebral region of the anterior aorta. The cephalic cavity is confluent with channels and lacunae supplying the jaws, mouth lips, and slime papillae. It posteriorly joins the perivisceral and the two lateral sinuses (LSI) of the trunk. The latter supply the lacunae and channels of each leg with hemolymph, which then passes via numerous slits through the bilateral transverse septa into the perivisceral sinus (Fig. 9b, c). From this, the hemolymph is then channeled into the pericardial sinus. This happens not only via the segmental pericardial channels but also via a posterior gap in the pericardial septum. The pericardial sinus dorsally receives additional hemolymph via dorsomedian channels from numerous subcutaneous, transverse, afferent ring channels (=plical channels) associated with the dermal plicae of the integument. During diastole, the hemolymph finally flows through the ostia from the pericardial sinus back into the lumen of the heart (Fig. 9a–c). According to our estimates, it might take the hemolymph 10 min and 45 seconds to circulate through the onychophoran body. We did not consider the observed posterior movement of some hemocytes in our calculations, as it is negligible and likely results from turbulences due to the influx via the (anterior) ostia during diastole. Moreover, there is no indication of actual backflow of hemolymph from the anterior aorta into the heart due to the presence of the cardiac valve between these two vessels. While the onychophoran heart seems to provide the major driving force for this circuit of hemolymph, we cannot exclude the potential involvement of the peristaltic movements of the gut and the body wall musculature in hemolymph circulation.

Besides this major circuit, there are additional channels and spaces that enable hemolymph circulation within each leg, in which the hydrostatic control seems to be independent from that in the remaining body cavity[56]. Our observations confirm that the connection between the lateral channels and each main leg cavity is interspersed with two proximal leg muscles that might regulate

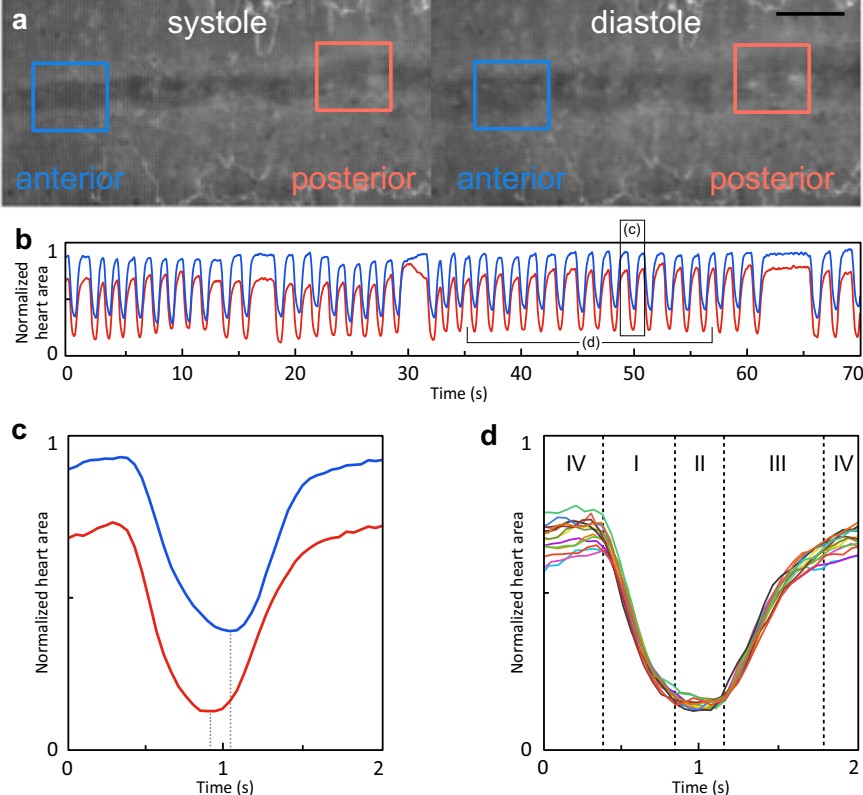

**Fig. 8 Heart contraction pattern in *E. rowelli*.** Light micrographs of live tissue (**a**); quantitative analysis of heart contractions (**b–d**). **a** Images of video recordings of systole/diastole. Colored squares indicate areas, in which contraction was measured. **b** Overview of contraction pattern. Colors correspond to areas indicated in **a**. Note repeated breaks at irregular intervals. **c** Corresponding contraction cycles in anterior and posterior areas indicated in a. Note nearly identical pattern, which is out of phase. **d** Overlay of 16 successive contractions reveal four different phases. Scale bar: 1 mm (**a**).

the hemolymph flow. The existence of transverse channels in each leg is also corroborated by our study. We assume that these channels as well as those associated with dermal plicae of the trunk and antennal rings might supply the integument and its sensory structures, including mechano- and chemoreceptors[92,103], with hemolymph. The direction of hemolymph flow in ring channels associated with legs and antennae is difficult to determine due to their small size and distant spatial relationship to the vascular and contractile structures. However, due to the lack of valves and specialized inflow/outflow openings, we assume that the hemolymph might be able to circulate in both directions in these channels.

Finally, the pulsatile organs are altogether missing in representatives of Tardigrada, the third major panarthropod clade. However, the presence of homologs of genes in the tardigrade genome that specify heart development in other panarthropods, such as *NK3* and *NK4*, suggests that a heart may have been lost in the tardigrade lineage due to the miniaturization of their body[10,11,13]. The same applies to other components of the circulatory system, which in extant tardigrades consists of a large, fluid-filled body cavity containing storage cells that are passively moved around during locomotion[11,12,94]. Irrespective of these potential losses in tardigrades, our findings suggest that the last common ancestor of Onychophora and Arthropoda exhibited an open vascular system (*sensu* Wirkner et al.[8]) with small vascular and relatively large lacunar parts. The heart was most likely situated in a pericardial sinus, which was incompletely separated from the perivisceral sinus by a pericardial septum with a large posterior gap. The wall of the heart was perforated by bilateral, slit-like segmental ostia associated with muscular valves that protruded deep into the cardiac lumen. Nephrocytic tissue, which

extends along the pericardial sinus on either side of the heart in both onychophorans and arthropods[5,9,57,104–106], might have constituted an efficient filter of solid waste matter before the hemolymph reached the heart[5,107]. The same applies to the nephridial system, which in extant onychophorans is associated with the two lateral sinuses and the main leg cavities[56,67,108–110] and is responsible for osmoregulation and excretion[5,111,112]. Hence, such a two-part filtration system of hemolymph, involving both nephrocytes and podocytes, might have also been present in the last common onychophoran-arthropod ancestor.

## Methods

**Specimens.** Specimens of *Euperipatoides rowelli* Reid, 1996 (Fig. 1) were obtained from leaf litter and rotten logs in Tallaganda State Forest, New South Wales, Australia (35°30′31″S, 149°36′14.3″E, 934 m). Animals were collected under the permit numbers SPPR0008 issued by the Forestry Commission of New South Wales and SL101720/2016 issued by the NSW National Parks and Wildlife Service and exported under the permit numbers WT2012-8163 and PWS2016-AU-001023 obtained from the Department of Sustainability, Environment, Water, Population and Communities. The animals were brought alive to the laboratory in Germany and kept in culture as described previously[113,114]. In total, 20 specimens were anesthetized using chloroform vapor prior to dissection or fixation. The experiments performed herein did not require approval by an ethics committee and all procedures fulfill the requirements of international and institutional guidelines, especially the guidelines for animal welfare, as recommended by the German Research Foundation (DFG).

**Video recordings.** One anesthetized specimen was dissected along the ventral midline and placed into a Petri dish filled with a 5-mm layer of polymerized Sylgard® (184 Silicone Elastomer Kit, DowCorning GmbH, Midland, MI, USA). A few droplets of physiological saline (Supplementary Protocol 1) based on the onychophoran blood composition[115] were placed on the dissected animal and the gut and slime glands were carefully pushed aside to expose the pumping heart. Live imaging videos (24 frames per second) were recorded under a stereomicroscope

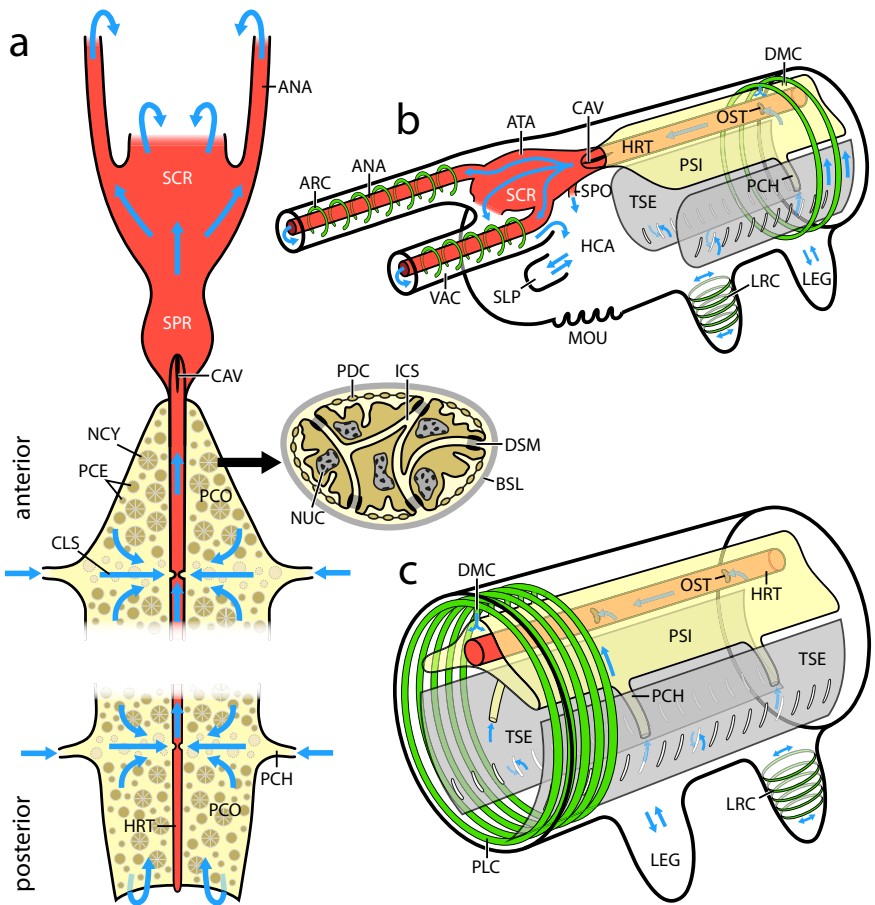

**Fig. 9 Diagrams of major circulatory pathways in *E. rowelli*.** For the sake of clarity, ventral antennal channels, main body cavity, lateral channels and leg cavities are not shown and only a few plical channels are illustrated. Flow directions of hemolymph are indicated by blue arrows. **a** Diagram of anterior aorta (SCR + SPR) and anterior and posterior parts of heart and pericardial sinus in dorsal view. Light shading of pericardial conglomerate indicates channel-like spaces that enable blood flow from pericardial channels via ostia into the lumen of heart. Inset illustrates details of a cluster of nephrocytes (after transmission electron microscopy data from a closely related species[57]). **b** Anterior body region in anterolateral perspective. **c** Diagram of two segments from midbody region in anterolateral perspective. Abbreviations: ANA antennal artery, ARC antennal ring channel, ATA anterior aorta, BSL basal lamina, CAV cardiac valve, CLS channel-like space between pericardial conglomerate, DMC dorsomedian channel, DSM desmosome, HCA head cavity, HRT heart, ICS intercellular space, LEG leg, LRC leg ring channel, MOU mouth, NCY nephrocytes, NUC nucleus, OST ostium, PCE pericardial cells, PCH pericardial channel, PCO pericardial conglomerate, PDC pedicels, PLC plical channel, PSI pericardial sinus, SCR supracerebral region of anterior aorta, SLP slime papilla, SPO ventral opening of suprapharyngeal region of anterior aorta, SPR suprapharyngeal region of anterior aorta, TSE transverse septum, VAC ventral antennal channel.

(Axio Zoom V16; Carl Zeiss Microscopy GmbH, Jena, Germany) equipped with a digital camera (Axiocam 503 color digital camera; Carl Zeiss Microscopy GmbH). Two regions at a distance of 1.5 mm were selected for measuring the peristaltic contractions of the heart. Exported videos were analyzed as image sequences with the open source software package Fiji[116,117]. After adjusting the binary thresholding, the number of white pixels in each slice of the image stack was determined using a compiled macro in Fiji (Supplementary Code 1). Obtained values were processed and graphs generated in Excel (Microsoft Corporation, Redmond, WA, USA). The measurements were normalized among 0–1 and depicted as their moving average using a sliding window of 20. Overlaid successive contractions were aligned at their lowermost values. After video recordings for heart rate measurements, the dorsal body wall containing the heart was segregated from the remaining body to investigate the ongoing contractions without a connection to the central nervous system.

**Calculation of physiological parameters**. Calculations of the physiological parameters of the heart were based on two data sets: (i) volume estimations resulting from three-dimensional reconstructions of one specimen (see 2.9.); and (ii) heart rate (*HR*) measurements of video recordings from another specimen (see 2.6.). The heart volume was calculated for two different conditions: end diastolic volume (*EDV*, $r_{max}$) and end systolic volume (*ESV*, $r_{min}$) with length of heart (l).

$$EDV = \pi * r_{max}^2 * l \qquad (5)$$

$$ESV = \pi * r_{min}^2 * l \qquad (6)$$

The stroke volume (*SV = EDV − ESV*), the ejection fraction (*EF = SV/EDV*) as well as the cardiac output (*CO = SV * HR*) were calculated.

**Scanning electron microscopy**. For SEM, five specimens were fixed either in glutaraldehyde (2.5% in 0.1 M sodium cacodylate buffer) for two hours and post-fixed overnight in $OsO_4$ (2% in 0.1 M phosphate-buffered saline [=PBS, Supplementary Protocol 2], pH 7.4) or in 4% paraformaldehyde (in PBS; Electron Microscopy Sciences, Hatfield, PA, USA) overnight. Samples were cut into smaller pieces using a razor blade, dehydrated in an increasing ethanol series, critical point dried in a CPD 030 (Bal-Tec AG, Balzers, Liechtenstein), glued onto standard aluminum stubs (diameter =12,5 mm; Plano GmbH, Wetzlar, Germany), sputter coated in a SCD 050 Sputter Coater (Balzers Union) and imaged using a field emission scanning electron microscope (Hitachi S-4000; Hitachi High-Technologies Europe GmbH, Krefeld, Germany) as described previously[118,119].

**Histology, semi-thin sectioning and light microscopy**. Two specimens prepared for histological sections were fixed with alcoholic Bouin's fluid (modified after Dubosq-Brasil[120,121]; Supplementary Protocol 3) and dehydrated in an increasing ethanol series followed by methyl benzoate and 1-butanol. After embedding in Paraplast® (Sherwood Medical Company, St. Louis, MO, USA) at 60 °C, specimens were cooled down and sectioned into 5-μm sections using a microtome (Leitz 1516;

Ernst Leitz GmbH, Wetzlar, Germany). The obtained sections were stained after the Heidenhain's Azan protocol[122] modified by Geidies[123]. For semi-thin sectioning, five specimens were fixed overnight in glutaraldehyde (2.5% in 0.1 M sodium cacodylate buffer) and post-fixed with 2% $OsO_4$ in PBS for 3 h. Thereafter, specimens were washed twice (30 min) in PBS, dehydrated in an increasing ethanol series, transferred into a 1:1 ethanol-acetone solution followed by two steps in 100% acetone, and embedded in Araldite® (Huntsman Advanced Materials, Salt Lake City, UT, USA) as described previously[20,108]. Semi-thin sections (0.5 μm) were cut with either glass or histo Jumbo diamond 45°, 6.0 mm (DiATOME, Science Services GmbH, Munich, Germany) knives on an Ultracut E ultramicrotome (Reichert-Jung, Optische Werke AG, Vienna, Austria) and stained with a staining solution containing methylene blue[124]. Histological and semi-thin sections were dried on a hot plate at 60 °C, mounted onto glass slides in Entellan® (Merck KGaA, Darmstadt, Germany) and analyzed with an Axio Imager M2 light microscope (Carl Zeiss Microscopy GmbH) equipped with a digital camera (Axiocam 503, Carl Zeiss Microscopy GmbH).

**Corrosion casting**. Injection experiments were based on protocols previously established for isopods[125], arachnids[32] and xiphosurans[23]. The casting resin (PU4ii kit, vasQtec, Zurich, Switzerland)[126] was injected into the heart of five specimens fixed in 4% PFA (in PBS) shortly before the experiment. Injection needles made from glass capillaries (Hirschmann Laborgeräte GmbH & Co. KG, Eberstadt, Germany) pulled in a DMZ-Universal-Electrode-Puller (Zeitz-Instrumente Vertriebs GmbH, Planegg, Germany) were used. The needles were filled with casting fluid (5 g of resin, 0.8 g of hardener, 3 g of Butan-2-one and 5–10 mg pigment powder) and mounted on an adjustable mechanical micromanipulator (Leitz, Wetzlar, Germany) connected to a 5-mL syringe (B. Braun Melsungen AG, Melsungen, Germany) via a silicone tube. The casting fluid was injected by gently pressing the mounted syringe. After polymerization overnight, the tissue was macerated for two days in 15% potassium hydroxide at room temperature followed by a digestion in 1 g pepsin in 50 mL of 2% HCL solution[127]. The samples were washed several times in water, air-dried and either imaged using a stereomicroscope (Axio Zoom V16; Carl Zeiss Microscopy GmbH) or prepared further for SEM analysis as described above.

**Immunohistochemistry and confocal laser scanning microscopy**. Dissected pieces of the dorsal body wall and heart from one specimen were fixed overnight in 4% paraformaldehyde (in PBS) at room temperature and rinsed three times in PBS. The samples were either used for whole-mount staining of tissues or embedded in 7% agarose (in PBS) at 60 °C, cooled down to room temperature, trimmed and sectioned into 60–80 μm thin sections on a vibratome (MICROM 650 V; MICROM International GmbH, Walldorf, Germany). Sections and whole-mount preparations were labeled for F-actin in a solution containing phalloidin-rhodamine (Thermo Fisher Scientific, Waltham, MA, USA) as described previously[128,129]. After several washing steps in PBS, the samples were mounted onto glass slides either in Vectashield Mounting Medium (Vector Laboratories Inc., Burlingame, CA, USA) or ProLong Gold Antifade Mountant (Thermo Fisher Scientific, Waltham, MA, USA) and analyzed under a confocal laser-scanning microscope (CLSM) Zeiss LSM 510 META or LSM 880 (Carl Zeiss Microscopy GmbH). The obtained image stacks were merged into final projections with the Zeiss LSM Image Browser software (Carl Zeiss Microscopy GmbH).

**Synchrotron radiation-based X-ray micro-computed tomography**. For SR-μCT, one specimen was initially fixed, contrasted and embedded as for semi-thin sectioning described above. SR-μCT scans were performed as described elsewhere[130]. The embedded specimen was glued on standardized sample holders (HZG, Geesthacht, Germany) using superglue (Pattex, Henkel AG & Co. KGaA, Düsseldorf, Germany) and analyzed at the beamline P05 of the storage ring PETRA III (Deutsches Elektronen-Synchrotron—DESY, Hamburg, Germany) operated by the Helmholtz-Zentrum Geesthacht[131,132] with photon energy of 23 keV for attenuation contrast[132]. A total of 900 radiograms were recorded at equal steps between 0 and π. The tomographic reconstruction algorithm "back-projection of filtered projections"[133] applied a binning factor of two and was used to yield 32-bit floating image stacks with an effective voxel size of 4.92 μm.

**Computer-based three-dimensional reconstruction**. Based on SR-μCT data from one individual, components of the circulatory system of *E. rowelli* were manually segmented and reconstructed three-dimensionally using Amira 6.0.1 (Thermo Fisher Scientific, Waltham, MA, USA). The main body cavity was automatically pre-segmented followed by manual fine adjustments. Each label field was smoothed by setting the pixel size to 3 in five consecutive smoothing steps followed by a double-check for relevant changes in general shape and arrangement by an overlay-comparison of smoothed labels with the SR-μCT data. Exported label fields were separated by applying the threshold function in Fiji[116]. Combined volume renderings of SR-μCT datasets and surface renderings of segmented parts of the circulatory system were generated in VG Studio MAX 3.0 (Volume Graphics GmbH, Heidelberg, Germany). Voxel number of each reconstructed part of the circulatory system was measured in VG Studio MAX 3.0. 3D reconstructions of ostia were based on digitalized images of serial semi-thin sections aligned in Amira

6.0.1 with subsequent reconstruction via threshold segmentation and surface rendering in VG Studio MAX 3.0. Exported 3D-reconstruction images as well as confocal, scanning electron and light micrographs were processed with Adobe Photoshop CS5.1 (Adobe Systems, San José, CA, USA). Final panels were compiled with Adobe Illustrator CS5.1.

**Terminology**. Whenever possible, the nomenclature for components of the circulatory system and associated structures in *E. rowelli* follows the terms and definitions established for arthropods (OArCS: Ontology of Arthropod Circulatory Systems)[83]. The original terminology of the OArCS database was amended and adapted for studies of Onychophora (Table 1 and Supplementary Table 1). The terminology and abbreviations for neural structures follow the nomenclature of Martin et al.[84].

**Reporting summary**. Further information on research design is available in the Nature Portfolio Reporting Summary linked to this article.

## Data availability

Data of this study are presented in the article and the Supplementary Information (Supplementary Code, Supplementary Data, Supplementary Figures, Supplementary Movies, Supplementary Protocols and Supplementary Tables). Source data of Fig. 8b–d are available in Supplementary Data 2. The image stack of the posteriorly blind ending heart and seven additional movies are available to download from figshare digital repository (https://doi.org/10.6084/m9.figshare.22233493). Further supporting data are available from the corresponding author upon reasonable request.

## Code availability

The code for counting white pixels of binary image stacks is available in the Supplementary Information. The presented code can be compiled as macro using the software Fiji[116].

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

## Acknowledgements

We are thankful to David M. Rowell and Noel N. Tait for their help with permits and to the members of the Mayer laboratory for their assistance with specimen collection and animal husbandry. HJ thanks Ivo de Sena Oliveira for assisting with macro photography and scanning electron microscopy and insightful discussions. Christine Martin and Vladimir Gross introduced HJ to confocal and scanning electron microscopes. Rick Hochberg kindly donated used diamond knives. Benjamin Tepper performed histological sectioning and Azan staining. Achim Werkenthin assisted with pulling glass capillaries for corrosion cast experiments. Ivo de Sena Oliveira and Andreas Kumerics provided scanning electron micrograph for Fig. 5f. Vladimir Gross provided linguistic suggestions on the manuscript. The staffs of the Forestry Commission of New South Wales and the Department of Sustainability, Environment, Water, Population and Communities are gratefully acknowledged for providing collecting and export permits. We acknowledge DESY (Hamburg, Germany), a member of the Helmholtz Association HGF, for the provision of experimental facilities. Parts of this research were carried out at PETRA III and we would like to thank for assistance in using Beamline P05. Beamtime was allocated for proposals I-20140177 and I-20150213 to GM and JUH. TG is supported by a postdoctoral fellowship from the German Research Foundation (Deutsche Forschungsgemeinschaft, DFG; GO 3341/1-1).

## Author contributions

H.J. and G.M. conceived study. H.J. performed experiments, collected data, designed figures and wrote first draft of manuscript. J.U.H. performed SR-μCT imaging and tomographic reconstruction. H.J., J.U.H., T.G., C.S.W. and G.M. analyzed data, discussed results and approved final manuscript.

## Funding

## Competing interests

The authors declare no competing interests.
