## [Peer Review File · Communications Biology]

Reviewers' comments:

Reviewer #1 (Remarks to the Author):

The present manuscript represents an accurate detailed study of circular system in Onychophora. It seems a very interesting and important piece of research to me, as it concerns a significant problem of Ecdysozoa phylogeny and evolution. The manuscript has a consistent and understandable structure, and is done at high quality level. As a (beginner) μ CT operator I am particularly impressed with elegant 3D reconstructions.

First of all I would like to notice, that I am an invertebrate morphologist myself, but my own research range is quite far from Panarthropoda. So my knowledge of Onychophora is rather incomplete; my comments and questions will concern mostly general issues, and might be somewhat naive. Besides, it is my first experience as a reviewer ever.

Comments on details:

1. There might be some ambiguity in descriptions of vessels' lining. Is there a true endothelium (epithelial tissue lining the inner surface of vessels) in Onychophora, or not? As far as I know, in most invertebrates (except e.g. nemerteans and leeches, vessels of which are actually derivatives of coelom) there is no true inner lining, and vessels are "lined" simply with basal lamina. You mention both endothelium (lines 49-50), and some non-epithelial lining (lines 265-266). Which of these statements is actual? If non-epithelial, what are these cells like?
2. Isn't it possible that chloroform anaesthesia could somehow affect heartbeat rate and amplitude?
3. Line 51: It will be probably be better to specify, that you use "hemocoel" and "myxocoel" as synonyms herein, as these terms are often understood differently (hemocoel as simply a primary cavity, and myxocoel as a fusion of primary and secondary cavities).
4. Lines 112-113: "...of tardigrades, onychophorans, arthropods and/or panarthropods". But panarthropods include all three mentioned groups; what does "and/or" mean in this case?
5. Lines 161-164: The abbreviations seem a bit intricate to me, especially V_{EDV} and V_{ESV} . Does the upper case V mean "ventricular"? If so, I doubt if it is suitable, as panarthropods do not have any ventricles or atria. If, on the other hand, this means "volume", then we have tautology: "volume end diastolic volume". I am not physiologist at all, and I could just misunderstand this part; so if these are common abbreviations required to use, please do not pay much attention to this comment.
6. Lines 259-263: The explanation of terminology is already given previously (lines 138-140); I suggest that it is more suitable for "Material and methods" and are not so necessary in "Results" part.
7. Lines 277-278: The "continuous cellular layer" is not obvious on presented figures. Saggital sections from μ CT stack cannot provide such an inference because of insufficient resolution, while semi-thin sections given in Fig. 3, Supl. Fig. 1, 3, 4 are too small. I think that a detailed photo of lining on a semi-thin section and/or a pair of TEM mini-images would be nice here.
8. Figure 3: "SPV" abbreviation is missed in the description.
9. Supplementary figure 1: "HRN" abbreviation is missing in the description.

Reviewer #2 (Remarks to the Author):

This is an impressive manuscript that takes a comprehensive approach to characterize in exquisite detail the circulatory system of velvet worms, placing the findings in the context of the evolution of the circulatory system of arthropods.

The manuscript is important in two primary ways. First, the comprehensive approach used – histology, histochemistry, confocal microscopy, scanning electron microscopy, synchrotron X-ray micro-

computed tomography, 3D reconstructions, and video recordings – provides what is probably the most detailed description of the circulatory system of an invertebrate in a single publishable unit. In this regard, the manuscript solidly delivers. Second, the manuscript provides insight into the evolution of the circulatory system of ecdysozoans. For this, the manuscript delivers the underlying argument although the level of detail in the presentation makes some of the arguments complicated to grasp because they are catered for the circulatory system specialist (good for the discipline) but less so to the generalist.

Overall, I am enthusiastic and supportive of this manuscript and do not have any significant concerns regarding the gross structural descriptions (in fact, they are outstanding). My primary concerns relate to the physiological recordings and some of the single cell descriptions.

1. Regarding physiological recordings. The data on physiological recordings gives me significant pause for several reasons. The experiments were conducted in immobilized dissected animals with added saline. The process of dissection and immersion (1) alters the pressure in the hemocoel and (2) removes all hemolymph factors that may be modifying the physiology of the heart. Hence, the recordings do not necessarily reflect what would be observed in an intact animal, including the heart rate and contraction strength. Moreover, the experimenters calculate the EDV and ESV under the assumption that the force of contraction is equal along the vessel, which is not likely the case. Also, in the absence of a complete constriction of the heart, the calculations of SV, EF and CO incorrectly assume that there is zero backflow; unlike for the vertebrate heart that contains valves that prevent backflow, these animals do not contain such valves. Finally, calculations on the amount of time that it takes for hemolymph to complete a cardiac cycle do not consider the role that other muscles in the body (including those used for movement) play in moving hemolymph. Finally, there is no mention of variance or sampling. So, between the effect of dissection, the assumptions for diastolic and systolic volume, the lack of consideration for other muscles, and the lack of mention of sampling and variance, I believe that these data are not representative (and quite possibly significantly off). It does not necessarily mean that they need to be entirely scrapped, but at least the caveats need to be presented, and the language tempered to reflect the uncertainty. (Elsewhere in the manuscript, quantitative descriptions on volume and area of different circulatory spaces are nicely done, although sampling and variance are missing there as well).

2. Regarding nephrocytes, hemocytes and other cells. The description on hemocytes is well done and the images convincing. My only comment is for the authors to acknowledge that hemocytes can also be attached to tissues (sessile). They may also be interested in that an infection recruits hemocytes to the heart of insects (I wonder whether this happens in velvet worms). I am less clear on the description of nephrocytes. I encourage the authors to more clearly define how nephrocytes are identified (how do they know it is this type of cell?). I also recommend noting that nephrocytes and pericardial cells are the same type of cell. This is noted in one of the extended tables, but it should also be clear in the text.

3. In the Conclusions section (and alluded elsewhere, like page 18), the authors called the cephalic aortas of onychophorans and arthropods as homologous whereas the posterior aorta is an innovation in arthropods. The argument does not seem to capture (or does not portray) that the posterior aorta may simply be a modification and extension of the cephalic aorta. After all, in both lineages the cephalic aorta and the heart are connected, either directly or by means of the posterior aorta.

4. I was unable to review the videos as they were not included in the submission.

Responses (in blue) to the reviewers' comments

Reviewer #1 (Remarks to the Author):

The present manuscript represents an accurate detailed study of circular system in Onychophora. It seems a very interesting and important piece of research to me, as it concerns a significant problem of Ecdysozoa phylogeny and evolution. The manuscript has a consistent and understandable structure, and is done at high quality level. As a (beginner) μ CT operator I am particularly impressed with elegant 3D reconstructions.

First of all I would like to notice, that I am an invertebrate morphologist myself, but my own research range is quite far from Panarthropoda. So my knowledge of Onychophora is rather incomplete; my comments and questions will concern mostly general issues, and might be somewhat naive. Besides, it is my first experience as a reviewer ever.

We thank Reviewer #1 for his/her positive comments and the many detailed and useful suggestions, which have helped to improve the manuscript.

Comments on details:

1. There might be some ambiguity in descriptions of vessels' lining. Is there a true endothelium (epithelial tissue lining the inner surface of vessels) in Onychophora, or not? As far as I know, in most invertebrates (except e.g. nemerteans and leeches, vessels of which are actually derivatives of coelom) there is no true inner lining, and vessels are "lined" simply with basal lamina. You mention both endothelium (lines 49-50), and some non-epithelial lining (lines 265-266). Which of these statements is actual? If non-epithelial, what are these cells like

Good point. We have modified the corresponding sentences in our Introduction as follows (line 50):

“The vascular systems, i.e. hearts and off-branching arteries of onychophorans and arthropods are characterized by cellular linings that are missing in their lacunar systems (Gaffron 1885; Nylund et al. 1988; Wirkner et al. 2013). These linings comprise non-epithelial, apolar cells (Mayer et al. 2015; Nylund et al. 1988; Rosenberg and Seifert 1978; Seifert and Rosenberg 1978) and in this respect they clearly differ from the vascular endothelium of vertebrates (Monahan-Earley et al. 2013). The body cavity of adult onychophorans comprises a hemocoel (sometimes referred to as mixocoel), which is surrounded by the extracellular matrix and arises during embryogenesis by mixocoely, i.e., a fusion of primary and secondary/coelomic body cavities (Mayer 2006; Mayer et al. 2004).”

2. Isn't it possible that chloroform anaesthesia could somehow affect heartbeat rate and amplitude?

We cannot exclude an effect of chloroform and have toned down our discussion in this regard.

Reviewer #2 had similar issues regarding the heart physiology. Please see our responses to point 1 of Reviewer #2 for physiological considerations.

3. Line 51: It will be probably be better to specify, that you use "hemocoel" and "myxocoel" as synonyms herein, as these terms are often understood differently (hemocoel as simply a primary cavity, and myxocoel as a fusion of primary and secondary cavities).

We have complemented this statement as follows (lines 53–57):

“The body cavity of adult onychophorans comprises a hemocoel (sometimes referred to as mixocoel), which is surrounded by the extracellular matrix and arises during embryogenesis by mixocoely, i.e., a fusion of primary and secondary/coelomic body cavities”.

We prefer not to use the term “mixocoel” but rather “mixocoely”, which refers to the fusion of primary and secondary/coelomic body cavities during embryogenesis in onychophorans and arthropods (Mayer 2006; Mayer et al. 2015; Mayer et al. 2004).

4. Lines 112-113: "...of tardigrades, onychophorans, arthropods and/or panarthropods". But panarthropods include all three mentioned groups; what does "and/or" mean in this case?

We have modified this sentence as follows (lines 117–121):

“Unfortunately, paleontological data from lobopodians, which most likely comprise a non-monophyletic assemblage of stem lineage representatives of tardigrades, onychophorans, arthropods, and panarthropods as a whole, respectively (Bergström and Hou 2001; Ortega-Hernández 2015; Ou and Mayer 2018), are not helpful for resolving these issues due to incomplete preservation and highly conjectural reconstructions of the circulatory system in these fossils (García-Bellido and Collins 2006; Göpel and Wirkner 2018; Liu et al. 2018; Ma et al. 2014).”

5. Lines 161-164: The abbreviations seem a bit intricate to me, especially V_{EDV} and V_{ESV} . Does the upper case V mean "ventricular"? If so, I doubt if it is suitable, as panarthropods do not have any ventricles or atria. If, on the other hand, this means "volume", then we have tautology: "volume end diastolic volume". I am not physiologist at all, and I could just misunderstand this part; so if these are common abbreviations required to use, please do not pay much attention to this comment.

Good point. This was a clear case of tautology. We have changed the abbreviations V_{EDV} and V_{ESV} to EDV and ESV throughout the manuscript (line 173 and line 344).

6. Lines 259-263: The explanation of terminology is already given previously (lines 138-140); I suggest that it is more suitable for "Material and methods" and are not so necessary in "Results" part.

This is also a very good point. We have included a new subsection entitled "2.10. Terminology" in our Materials and Methods (line 266–271), to which we have moved all other sentences dealing with morphological terminology.

7. Lines 277-278: The "continuous cellular layer" is not obvious on presented figures. Saggital sections from μ CT stack cannot provide such an inference because of insufficient resolution, while semi-thin sections given in Fig. 3, Supl. Fig. 1, 3, 4 are too small. I think that a detailed photo of lining on a semi-thin section and/or a pair of TEM mini-images would be nice here.

For the sake of clarity, we have included arrows in Supplementary Figure 2a pointing to the continuous layer of tissue, which lines the lumen of the anterior aorta. We have added the following sentence to the corresponding figure legend (line 1300):

"Note that lumen of anterior aorta is lined by continuous layer of tissue (arrows)."

8. Figure 3: "SPV" abbreviation is missed in the description.

We have changed "SPV" into "SPR" and updated this change in Figure 3c.

9. Supplementary figure 1: "HRN" abbreviation is missing in the description.

We have changed "HRN" into "HAN" and updated this change in Supplementary Figure 1a.

Reviewer #2 (Remarks to the Author):

This is an impressive manuscript that takes a comprehensive approach to characterize in exquisite detail the circulatory system of velvet worms, placing the findings in the context of the evolution of the circulatory system of arthropods.

The manuscript is important in two primary ways. First, the comprehensive approach used – histology, histochemistry, confocal microscopy, scanning electron microscopy, synchrotron X-ray micro-computed tomography, 3D reconstructions, and video recordings – provides what is probably the most detailed description of the circulatory system of an invertebrate in a single publishable unit. In this regard, the manuscript

solidly delivers. Second, the manuscript provides insight into the evolution of the circulatory system of ecdysozoans. For this, the manuscript delivers the underlying argument although the level of detail in the presentation makes some of the arguments complicated to grasp because they are catered for the circulatory system specialist (good for the discipline) but less so to the generalist.

Overall, I am enthusiastic and supportive of this manuscript and do not have any significant concerns regarding the gross structural descriptions (in fact, they are outstanding). My primary concerns relate to the physiological recordings and some of the single cell descriptions.

We thank Reviewer #2 for his/her critical and helpful comments that we have tried to address the best we could. We hope that our changes satisfy the reviewer.

1. Regarding physiological recordings. The data on physiological recordings gives me significant pause for several reasons. The experiments were conducted in immobilized dissected animals with added saline. The process of dissection and immersion (1) alters the pressure in the hemocoel and (2) removes all hemolymph factors that may be modifying the physiology of the heart. Hence, the recordings do not necessarily reflect what would be observed in an intact animal, including the heart rate and contraction strength. Moreover, the experimenters calculate the EDV and ESV under the assumption that the force of contraction is equal along the vessel, which is not likely the case. Also, in the absence of a complete constriction of the heart, the calculations of SV, EF and CO incorrectly assume that there is zero backflow; unlike for the vertebrate heart that contains valves that prevent backflow, these animals do not contain such valves. Finally, calculations on the amount of time that it takes for hemolymph to complete a cardiac cycle do not consider the role that other muscles in the body (including those used for movement) play in moving hemolymph. Finally, there is no mention of variance or sampling. So, between the effect of dissection, the assumptions for diastolic and systolic volume, the lack of consideration for other muscles, and the lack of mention of sampling and variance, I believe that these data are not representative (and quite possibly significantly off). It does not necessarily mean that they need to be entirely scrapped, but at least the caveats need to be presented, and the language tempered to reflect the uncertainty. (Elsewhere in the manuscript, quantitative descriptions on volume and area of different circulatory spaces are nicely done, although sampling and variance are missing there as well).

To address the issue of sampling size, we have added the information on the number of investigated specimens to each corresponding subsection of our Materials & Methods. In total, we have analyzed 20 individuals of *E. rowelli*, which is a relatively high number, given that specimens of Onychophora are difficult to obtain.

We agree that the artificial conditions might have affected the outcome of our physiological experiments. However, the results of our video recordings strikingly correspond to previous measurements of heart beat rate (Sundara Rajulu and Singh 1969) and electrocardiograms combined with intracellular recordings (Hertel et al. 2002) in two distantly related onychophoran species. In these studies, even more invasive methods were applied, as the authors investigated isolated hearts or heart segments that were placed in saline. Sundara Rajulu and Singh (1969) used the same Robson's saline as applied in our study after Robson et al. (1966), whereas Hertel et al. (2002) utilized "normal saline" after Cook and Holman (1975). From both studies, nothing is known about their anesthetization and dissection techniques. We preferred anesthetization prior to dissection in our experiments for ethical reasons. After gentle opening the animal ventrally, the gut was carefully pushed aside and video recordings of the pumping heart were carried out. Compared to the mentioned previous studies, our approach was much less invasive to analyze the pumping activity of the heart.

As pointed out above, our results strikingly correspond with those from the two previous studies. The only difference is that neither Sundara Rajulu and Singh (1969) nor Hertel et al. (2002) detected intermittent heartbeat, probably because their recordings lasted for a much shorter period of time. This indicates that our estimates might not be "significantly off" the real circumstances. Unfortunately, *in vivo* measurements of heartbeat are currently infeasible in onychophorans and it would be difficult to obtain a sufficient number (several dozens) of specimens for analyzing the effects of various agents and components of blood or physiological saline on the heartbeat rate. As described in section 3.1, the heart exhibits a peristaltic contraction pattern leading in anterior direction, which – in our opinion – renders backflow within the heart lumen insignificant. Backflow into the heart during diastole is actually prevented by a valve between the heart and its only outlet, the anterior aorta. With these aspects in mind, we do not assume significant backflow.

We have toned down the manuscript with respect to the quantitative calculations of blood flow. We rewrote the Discussion as follows (lines 607–615):

"While we are aware that immobilization and dissection of specimens and application of saline might alter the physiological state of the individual, the observed regular heartbeat in *E. rowelli* and its striking correspondences to the results from two other onychophoran species based on entirely different datasets (heart beat rate measurements by Sundara Rajulu and Singh, 1969; electrocardiograms and intracellular recordings by Hertel et al., 2002) indicate that these measurements might be close to the natural conditions. All these studies indicate that the heart nerve might control the rhythmicity of contractions autonomously without an input from the central nervous system."

We did not provide hemolymph pressure data, which of course would be completely different in the intact animal. We only provide metrics of the heart, although they certainly are also regulated by hemolymph parameters. That being said, the mere metrics in the relaxed and constricted state of the heart are likely less off than, e.g., hemolymph pressure or flow velocity would be.

We agree that other muscles of the onychophoran body might influence the circulation of hemolymph in lacunar spaces. However, all hemolymph passes through the heart. Thus, the cardiac output (=CO) is the bottleneck defining how much hemolymph can be circulated in a certain time. The limitations regarding the validity/representative power of our calculation therefore stem only from the artificial physiological state under which CO was determined. Not, however, from the fact that the calculation is based on CO only.

We hope that our changes and amendments are sufficient and satisfy the reviewer.

2. Regarding nephrocytes, hemocytes and other cells. The description on hemocytes is well done and the images convincing. My only comment is for the authors to acknowledge that hemocytes can also be attached to tissues (sessile). They may also be interested in that an infection recruits hemocytes to the heart of insects (I wonder whether this happens in velvet worms). I am less clear on the description of nephrocytes. I encourage the authors to more clearly define how nephrocytes are identified (how do they know it is this type of cell?). I also recommend noting that nephrocytes and pericardial cells are the same type of cell. This is noted in one of the extended tables, but it should also be clear in the text.

Thanks for pointing this out. We have now specified in line 461 and in Table 1 that the hemocytes “are either sessile (attached to various tissues) or float freely in the hemolymph”. We are aware of the fact that hemocytes might be part of the hematopoietic or immune system in insects, but we do not know anything about the function of these cells in onychophorans. Interestingly, the pigment-dispersing factor neuropeptide receptor (PDFR) is expressed in the membrane of hemocytes found in the body cavity of *E. rowelli* (cf. fig. 9C in Martin et al. 2022). We also have some interesting unpublished data on other aspects of these cells, but including these data would be beyond the scope of our present manuscript.

Fig. 10: Diagram of a nephrocyte included in a.

We have included additional text on p. 16 (line 465–471) and an additional diagram (Fig. 10a), which might help to clarify, what nephrocytes are. Please note that beyond nephrocytes, the pericardial sinus contains other, hemocyte-like cells, which is why we prefer to use the term “pericardial cells” for both. Therefore, “pericardial cells” and “nephrocytes” are not synonyms. Inconsistencies in the literature are obvious, as Manton and Heatley (1937) used the term “nephrocyte” for the small uninucleate cell types and the term “pericardial cell” for the cell clusters. Therefore, we decided to choose the expression “uncharacterized pericardial cells” for these cells in our text (Results 3.3; p. 16, line 479). Additionally, we included the following definition for pericardial cells in Table 1:

“Small, uncharacterized cells, which together with nephrocytes form longitudinal bands on either side of →heart within →pericardial sinus”.

We have further added the label “PCE, pericardial cell” to Figure 8e and Supplementary Figures 4f and 5c,d. The cell types within the pericardial sinus would definitely require further investigation.

3. In the Conclusions section (and alluded elsewhere, like page 18), the authors called the cephalic aortas of onychophorans and arthropods as homologous whereas the posterior aorta is an innovation in arthropods. The argument does not seem to capture (or does not portray) that the posterior aorta may simply be a modification and extension of the cephalic aorta. After all, in both lineages the cephalic aorta and the heart are connected, either directly or by means of the posterior aorta.

The anterior and posterior aortas are distinct structures. While the anterior aortas of onychophorans and arthropods are situated anterior to the heart, the posterior aortas of (some) arthropods are located posterior to the heart, as illustrated in the following diagram from Wirkner et al. (2013):

The absence of a posterior aorta in onychophorans and its presence only in a few distantly related arthropod species indicates that this vessel might have evolved independently in several arthropod lineages. We therefore believe that our conclusions in this respect are well justified. We have replaced the term “cephalic aorta” with “anterior aorta”, which were used as synonyms in our previous version, throughout the manuscript to avoid confusion. Furthermore, we have included information about the different positions of the anterior aortas in onychophorans and arthropods on p. 18 (lines 550–556) of our revised Discussion, which now reads as follows:

“The position of the anterior aorta relative to the brain is also different in both groups. In arthropods, it passes the brain ventrally, whereas it does so dorsally in onychophorans (Pass 1991).” and further: “an anterior aorta might have been present in the last common ancestor of Onychophora and Arthropoda, but its shape and relative position to the brain might have been changed in either lineage. This is not surprising, given the extensive and most likely independent reorganizations of the head during cephalization processes in different panarthropod lineages (Budd 2002; Martin et al. 2022; Ortega-Hernández et al.

2017; Ou et al. 2012; Scholtz and Edgecombe 2006; Scholtz and Edgecombe 2005)".

4. I was unable to review the videos as they were not included in the submission.

We are sorry for the inconvenience. Please find the videos under the link:

<https://hessenbox.uni-kassel.de/getlink/fiJmRFNpBcfptgNFpSx4T57c/videos>

References:

- Bergström J, Hou XG (2001) Cambrian Onychophora or Xenusians. *Zool Anz* 240:237–245.
- Budd GE (2002) A palaeontological solution to the arthropod head problem. *Nature* 417:271–275.
- Cook BJ, Holman GM (1975) Sites of action of a peptide neurohormone that controls hindgut muscle activity in the cockroach *Leucophaea maderae*. *J Insect Physiol* 21:1187–1192.
- Gaffron E (1885) Beiträge zur Anatomie und Histologie von *Peripatus*. *Zool Beitr* 1:33–60.
- García-Bellido DC, Collins DH (2006) A new study of *Marrella splendens* (Arthropoda, Marrellomorpha) from the Middle Cambrian Burgess Shale, British Columbia, Canada. *Can J Earth Sci* 43:721–742.
- Göpel T, Wirkner CS (2018) Morphological description, character conceptualization and the reconstruction of ancestral states exemplified by the evolution of arthropod hearts. *PLOS ONE* 13:e0201702.
- Hertel W, Wirkner CS, Pass G (2002) Studies on the cardiac physiology of Onychophora and Chilopoda. *Comp Biochem Phys A* 133:605–609.
- Liu J, Steiner M, Dunlop JA, Shu D (2018) Microbial decay analysis challenges interpretation of putative organ systems in Cambrian fuxianhuids. *Proc R Soc Lond B Biol Sci* 285.
- Ma X, Cong P, Hou X, Edgecombe GD, Strausfeld NJ (2014) An exceptionally preserved arthropod cardiovascular system from the early Cambrian. *Nat Commun* 5:3560.
- Manton SM, Heatley NG (1937) Studies on the Onychophora. II. The feeding, digestion, excretion, and food storage of *Peripatopsis*, with biochemical estimations and analyses. *Phil Trans R Soc Lond B Biol Sci* 227:411–464.
- Martin C, Jahn H, Klein M, Hammel JU, Stevenson PA, Homberg U, Mayer G (2022) The velvet worm brain unveils homologies and evolutionary novelties across panarthropods. *BMC Biol* 20:26.
- Mayer G (2006) Origin and differentiation of nephridia in the Onychophora provide no support for the Articulata. *Zoomorphology* 125:1–12.
- Mayer G, Franke FA, Treffkorn S, Gross V, Oliveira IS (2015) Onychophora. In: Wanninger A (ed) *Evolutionary developmental biology of invertebrates 3: Ecdysozoa I: Non-tetraconata*. Springer, Vienna, pp 53–98.
- Mayer G, Ruhberg H, Bartolomaeus T (2004) When an epithelium ceases to exist — An ultrastructural study on the fate of the embryonic coelom in *Epiperipatus biolleyi* (Onychophora, Peripatidae). *Acta Zool* 85:163–170.
- Monahan-Earley R, Dvorak AM, Aird WC (2013) Evolutionary origins of the blood vascular system and endothelium. *J Thromb Haemost* 11:46–66.
- Nylund A, Ruhberg H, Tjonneland A, Meidell B (1988) Heart ultrastructure in four species of Onychophora (Peripatopsidae and Peripatidae) and phylogenetic implications. *Zool Beitr* 32:17–30.
- Ortega-Hernández J (2015) Lobopodians. *Curr Biol* 25:R873–R875.
- Ortega-Hernández J, Janssen R, Budd GE (2017) Origin and evolution of the panarthropod head—A palaeobiological and developmental perspective. *Arthropod Struct Dev* 46:354–379.

- Ou Q, Mayer G (2018) A Cambrian unarmoured lobopodian, †*Lenisambulatrix humboldti* gen. et sp. nov., compared with new material of †*Diania cactiformis*. *Sci Rep* 8:13667.
- Ou Q, Shu D, Mayer G (2012) Cambrian lobopodians and extant onychophorans provide new insights into early cephalization in Panarthropoda. *Nat Commun* 3.
- Pass G (1991) Antennal circulatory organs in Onychophora, Myriapoda and Hexapoda: Functional morphology and evolutionary implications. *Zoomorphology* 110:145–164.
- Robson EA, Lockwood APM, Ralph R (1966) Composition of the blood in Onychophora. *Nature* 209:533.
- Rosenberg J, Seifert G (1978) Feinstruktur der Innervierung des Dorsalgefäßes von *Peripatoides leuckarti* (Saenger 1869) (Onychophora, Peripatopsidae). *Zool Anz* 201:21–30.
- Scholtz G, Edgecombe GD (2005) Heads, Hox and the phylogenetic position of trilobites. In: Koenemann S, Jenner RA (eds) *Crustacea and Arthropod Relationship*, vol 16. CRC Press, Boca Raton, Florida, pp 139–165.
- Scholtz G, Edgecombe GD (2006) The evolution of arthropod heads: reconciling morphological, developmental and palaeontological evidence. *Dev Genes Evol* 216:395–415.
- Seifert G, Rosenberg J (1978) Feinstruktur der Herzwand des Doppelfüßers *Oxidus gracillis* (Diplopoda: Paradoxosomatidae) und allgemeine Betrachtungen zum Aufbau der Gefäße von Tracheata und Onychophora. *Entomol Ger*:224–233.
- Sundara Rajulu G, Singh M (1969) Physiology of the heart of *Eoperipatus weldoni* (Onychophora). *Naturwissenschaften* 56:38.
- Wirkner CS, Tögel M, Pass G (2013) The arthropod circulatory system. In: Minelli A, Boxshall G, Fusco G (eds) *Arthropod biology and evolution: Molecules, development, morphology*. Springer Berlin Heidelberg, Berlin, Heidelberg, pp 343–391.

Reviewers' comments:

Reviewer #2 (Remarks to the Author):

This revised manuscript improves an already strong manuscript, so like for the initial submission, I am supportive of this manuscript. In my initial review I outlined the reasons why this research is impactful, so I will not repeat myself here.

In my initial review I had three main points.

The second point pertained to the description of nephrocytes, hemocytes and other cells. The authors satisfactorily addressed this comment.

The third point pertained to the anterior versus posterior aorta. The authors present a clear explanation, and my comment originated from a misreading on my part because in insects the posterior region of the aorta (insects only have an anterior aorta) is often called the posterior aorta.

The first point—which was most important—pertained to the physiological measurements. In my argument, I recommended that the authors present the caveats of their experiment. I did so, because I have worked on an organism whose heart physiology changes when the organism is dissected relative to when it is intact. The authors' response to my comment has been to double down on their argument. Here I will explain some of the problems with their rebuttal logic. For ease, I will number them.

1. If I read the methods correctly, these measurements were only made in two individuals, so even if the method and recordings were perfect, it is unclear whether the values are representative because of the low sampling. At the least, the values for both individuals should be included. I understand that specimen collection is difficult, so I am not asking for more sampling. However, this caveat should be noted.

2. The authors postulate that their method is much better than prior methods (Sundara; Hertel) so their measurements should be more accurate, yet use the concordance between their values and those of the other two studies to support the physiological accuracy of their recordings. Given that the methods in all three studies involved dissection, removal of hemolymph components, disruption of hemolymph pressure, etc., their concordance cannot be used to claim that what is seen is what would be seen in an intact specimen. Again, mentioning this caveat would be sufficient.

3. I also mentioned that the authors are not accounting for any possible backflow, and the authors counter by writing in the rebuttal letter "the heart exhibits a peristaltic contraction pattern leading in anterior direction, which – in our opinion – renders backflow within the heart lumen insignificant. Backflow into the heart during diastole is actually prevented by a valve between the heart and its only outlet, the anterior aorta." This statement appears contradicted in the revised manuscript, where the authors write in line 335, "After a fast forward flow of hemolymph during the systole, it subsequently slows down and the hemocytes indicate a short backflow during the diastole". That backflow is precisely what I was referring to.

4. Finally, the authors dismiss the potential impact of pressure in the hemocoel. I don't think this is wise. If the heart is always propelling hemolymph anteriorly, one would expect hemocoelic pressure to be highest in the anterior of the organism, which would provide some resistance to cardiac/aorta flow. When the animal is dissected, that pressure differential is eliminated.

To summarize, this is a strong and meaningful study, but without clearly noting the potential caveats of the physiological data this portion of the manuscript could be misleading. I recommend that the authors state the caveats of their experiments. It is a simple fix.

Reviewer #3 (Remarks to the Author):

"This paper has been reviewed previously and a full review is not requested at this stage. However, we would appreciate your input on assessing the authors responses to Reviewer 1"

Reviewer 1's comments were sensible and straightforward, and I feel the authors have addressed them satisfactorily.

Responses (in blue) to the reviewers' comments

Reviewer #2 (Remarks to the Author):

This revised manuscript improves an already strong manuscript, so like for the initial submission, I am supportive of this manuscript. In my initial review I outlined the reasons why this research is impactful, so I will not repeat myself here.

In my initial review I had three main points.

The second point pertained to the description of nephrocytes, hemocytes and other cells. The authors satisfactorily addressed this comment.

The third point pertained to the anterior versus posterior aorta. The authors present a clear explanation, and my comment originated from a misreading on my part because in insects the posterior region of the aorta (insects only have an anterior aorta) is often called the posterior aorta.

The first point—which was most important—pertained to the physiological measurements. In my argument, I recommended that the authors present the caveats of their experiment. I did so, because I have worked on an organism whose heart physiology changes when the organism is dissected relative to when it is intact. The authors' response to my comment has been to double down on their argument. Here I will explain some of the problems with their rebuttal logic. For ease, I will number them.

We thank Reviewer #2 again for his/her helpful comments on our manuscript. Following his/her criticisms regarding our physiological measurements, we have modified the respective parts of the manuscript and hope that our amendments now satisfy the Reviewer.

1. If I read the methods correctly, these measurements were only made in two individuals, so even if the method and recordings were perfect, it is unclear whether the values are representative because of the low sampling. At the least, the values for both individuals should be included. I understand that specimen collection is difficult, so I am not asking for more sampling. However, this caveat should be noted.

We admit that our description was ambiguous, as we actually used different data sets from each of the two individuals for our calculations. We have now specified this in our revised Methods section (p. 6) as follows:

“Calculations of the physiological parameters of the heart were based on two data sets: (i) volume estimations resulting from three-dimensional reconstructions of one specimen (see 2.9.); and (ii) heart rate (*HR*) measurements of video recordings from another specimen (see 2.6.).”

The corresponding data are included in the manuscript.

2. The authors postulate that their method is much better than prior methods (Sundara; Hertel) so their measurements should be more accurate, yet use the concordance between their values and those of the other two studies to support the physiological accuracy of their recordings. Given that the methods in all three studies involved dissection, removal of hemolymph components, disruption of hemolymph pressure, etc., their concordance cannot be used to claim that what is seen is what would be seen in an intact specimen. Again, mentioning this caveat would be sufficient.

We have not postulated that our method is much better. We only wrote that we have measured for a longer period of time than in previous studies, which is why previous authors might have been unable to detect the intermittent contractions of the heart (p. 19):

“Quantitative measurements revealed that heart contractions occur periodically in *E. rowelli* in anteriorly directed peristaltic waves. The *in situ* measured heart rate is in line with the results of physiological studies in other species, such as *Eoperipatus weldoni* and *Peripatopsis* sp., except that in *E. rowelli* we additionally observed regular breaks after several successive rounds of contraction. These intermittent contractions might have remained unnoticed in onychophorans because their detection requires a prolonged monitoring of heart rate.”

On p. 20 of our Discussion we have elaborated further:

“While we are aware that immobilization and dissection of specimens and application of saline might alter the hemolymph pressure as well as the physiological state of the individual, the observed regular heartbeat in *E. rowelli* and its striking correspondences to the results from two other onychophoran species based on entirely different datasets and methodologies (heart beat rate measurements, electrocardiograms and intracellular recordings) indicate that these measurements might be close to the natural condition.”

We have now added the following sentence to this paragraph:

“However, we cannot rule out that there might be also an input from the central nervous system in intact specimens, as all three studies involved dissection, removal of hemolymph components, and disruption of hemolymph pressure, which might have affected heart function.”

We believe that we are now cautious enough and hope that these amendments satisfy the Reviewer.

3. I also mentioned that the authors are not accounting for any possible backflow, and the authors counter by writing in the rebuttal letter “the heart exhibits a peristaltic contraction pattern leading in anterior direction, which – in our opinion – renders backflow within the heart lumen insignificant. Backflow into the heart during diastole is actually prevented by a valve between the heart and its only outlet, the anterior

aorta.” This statement appears contradicted in the revised manuscript, where the authors write in line 335, “After a fast forward flow of hemolymph during the systole, it subsequently slows down and the hemocytes indicate a short backflow during the diastole”. That backflow is precisely what I was referring to.

There still seems to be a misunderstanding. The observed backflow of hemocytes does not indicate a backflow of hemolymph from the anterior aorta back into the heart lumen, but might rather have resulted from the influx of hemolymph via the further anteriorly situated ostia. It thus occurs in the non-ejected fraction of hemolymph in the heart and does not alter overall hemolymph flow in the body. Moreover, the observed backflow is negligible as compared to the fast and extensive forward propulsion of hemocytes. We have modified our text in several places to make these points clearer as follows.

P. 11 of Results:

“The forward movement of the hemocytes during the systole is rapid, so that they are hardly detectable at the original speed of the recording, whereas the posterior movement during the diastole is considerably slower and occurs over a much shorter distance.”

P. 17 of Results:

“Our data suggest that the heart of *E. rowelli* releases hemolymph into the anterior aorta via an anteroventral slit, the cardiac valve, which most likely prevents a reflux of hemolymph back into the heart.”

P. 21 of Discussion:

“The ostia and the cardiac valve ensure overall unidirectional circulation.”

And further:

“We did not consider the observed posterior movement of some hemocytes in our calculations, as it is negligible and likely results from turbulences due to the influx via the (anterior) ostia during diastole. Moreover, there is no indication of actual backflow of hemolymph from the anterior aorta into the heart due to the presence of the cardiac valve between these two vessels.”

4. Finally, the authors dismiss the potential impact of pressure in the hemocoel. I don't think this is wise. If the heart is always propelling hemolymph anteriorly, one would expect hemocoelic pressure to be highest in the anterior of the organism, which would provide some resistance to cardiac/aorta flow. When the animal is dissected, that pressure differential is eliminated.

We agree that the hemolymph pressure might be eliminated in a dissected specimen, which is in fact why we did not include this parameter in our

physiological considerations. We have specified on p. 20 of our Discussion that the potential changes of the hemolymph pressure due to dissection might have affected the outcome of our measurements. We refrain from discussing the issue of potentially different hemolymph pressures along the onychophoran body, as we do not have the corresponding data and would like to avoid too much speculation.

To summarize, this is a strong and meaningful study, but without clearly noting the potential caveats of the physiological data this portion of the manuscript could be misleading. I recommend that the authors state the caveats of their experiments. It is a simple fix.

We are thankful for the positive and helpful comments and hope that we could eliminate most of the shortcomings from the manuscript.

Reviewer #3 (Remarks to the Author):

"This paper has been reviewed previously and a full review is not requested at this stage. However, we would appreciate your input on assessing the authors responses to Reviewer 1"

Reviewer 1's comments were sensible and straightforward, and I feel the authors have addressed them satisfactorily.

We thank Reviewer #3 for his/her positive recommendation.

REVIEWERS' COMMENTS:

Reviewer #2 (Remarks to the Author):

The authors have addressed my comments. This is a strong manuscript.